# Global distribution of research efforts, disease burden, and impact of US public funding withdrawal

Leo Schmallenbach [1], Maximilian Bley [2,3], Till W. Bärnighausen [4,5,6], Cassidy R. Sugimoto [7,8], Carolin Lerchenmüller [2,3,9,10] & Marc J. Lerchenmueller [1,11] ✉

Evaluating whether research aligns with the global burden of disease is essential for equitable and effective scientific progress and improvement of human health. Without systematic evaluation of this alignment, science cannot respond to shifting health needs. Here we analyzed the distribution between research and disease, linking 8.6 million disease-specific publications to two decades of global disease burden data using a triangulated large language model approach. We find that since 1999, research and disease burden have seemingly become much more aligned; however, this is mainly because of regional declines in communicable disease burden, whereas the noncommunicable disease burden has increased and globalized. Meanwhile, research effort has not changed to match changes in disease burden. Our simulations suggest that without intentional alignment, the research–disease divergence will probably widen by a third over the next two decades, and be substantially accelerated by the reduction of US public funding for international research. Aligning research with health needs will require strategic investments, improved global coordination, open science policies and stronger, more equitable international partnerships to build resilience in a fragile research ecosystem.

Global health, commonly assessed in terms of disability-adjusted life years (DALYs), has improved in recent decades[1,2]. Yet, concerns prevail that the research enterprise continues to diverge from global health priorities, suggesting untapped potential for health improvements. Cross-sectional studies have documented a stark divergence at several points in time between the distribution of research efforts on specific diseases and the corresponding disease burden at national and international levels[3–10]. This divergence challenges a central tenet of post-World War II science policy: research is a public good that should contribute to the well-being of society[11]. As such, science requires public funding[11,12]; the social contract underlying the rationale of this funding is that the research is responsive to societal needs[13,14]. For example, the doubling of the budget for the National Institutes of Health in the United States between 1998 and 2002 was justified on the promise of health benefits for Americans[15] and has been shown to also improve health-related wealth, including patents generated, drugs discovered and creation of jobs[16–20].

[1]University of Mannheim, Mannheim, Germany. [2]Chair for Gender Medicine, University of Zurich, Zurich, Switzerland. [3]Department of Cardiology, University Hospital Zurich, Zurich, Switzerland. [4]Heidelberg Institute of Global Health (HIGH), Medical School, Heidelberg University, Heidelberg, Germany. [5]Harvard Center for Population and Development Studies and Harvard T.H. Chan School of Public Health, Harvard University, Cambridge, MA, USA. [6]Africa Health Research Institute (AHRI), Durban, South Africa. [7]School of Public Policy, Georgia Institute of Technology, Atlanta, GA, USA. [8]Centre for Research on Evaluation, Science and Technology (CREST), Stellenbosch University, Stellenbosch, South Africa. [9]Department of Cardiology, University Hospital Heidelberg, Heidelberg, Germany. [10]German Center for Cardiovascular Research (DZHK), Partner Site Heidelberg/Mannheim, Heidelberg, Germany. [11]Leibniz Center for European Economic Research, Mannheim, Germany. ✉e-mail: marc.lerchenmueller@uni-mannheim.de

**Table 1 | Summary of key findings and implications for policy**

| Background | Evaluating whether research aligns with the global burden of disease (GBD) is essential for equitable and effective scientific progress and improvement of human health. Tracking research–disease alignment over time reveals whether research priorities are adjusted to shifting health needs or reinforce persistent gaps. This study takes a global, longitudinal view to identify neglected areas, evolving trends and opportunities to shape a more responsive, inclusive research enterprise and evidence-based science policy. |
|---|---|
| Main findings and limitations | We use a triangulated approach, combining an LLM, international classification of disease codes and medical expert validation, to reveal that the divergence between research and disease burden has narrowed by 50% over the past two decades. This narrowing has been driven almost entirely by a shift in the GBD. We find a dichotomy of more local communicable diseases, which have seen a decline in divergence of about 75%, and more global noncommunicable diseases, which have seen a 25% increase in divergence, together yielding a halving (−75% + 25%) of the research–disease divergence since 1999. A forecasting of the divergence to 2050 shows that if this dichotomy is not addressed with a more aligned research enterprise, the divergence is likely to widen again. Accounting for a withdrawal of US public funding for international research would sharply accelerate a future widening of the divergence. The simulation of the future is limited by the inherent fragility of the global research enterprise. |
| Policy implications | The identified dichotomy calls for differentiated approaches:<br>• Global noncommunicable diseases affect populations worldwide, but most research is still concentrated in research-intensive countries. Policy action, like open science and data sharing mandates, is needed to make progress in aligning global research efforts with evolving disease burden. This will facilitate the acceleration of research in traditionally less research-intensive countries, which become increasingly affected by the burden of noncommunicable diseases.<br>• For more localized communicable diseases, it would be required to make a concerted effort to invest in locally led research capacity and equitable collaboration frameworks that avoid extractive practices (helicopter science) and build sustainable, research-informed responses to localized diseases.<br>In general, to more deliberately address research–disease divergences going forward, our findings underscore the importance of real-time monitoring to devise responsive funding strategies that should be coordinated by global bodies. This coordination should be strengthened through permanent governance mechanisms, especially secured funding and treaty-based commitments. Key research countries are integral to these efforts, yet a changing global policy and research landscape may jeopardize progress. For example: a potential withdrawal of US public funding for international research would affect communicable diseases such as HIV/AIDS, respiratory diseases and tuberculosis, as well as noncommunicable diseases, such as neurological diseases and substance use disorders. In the short term, countries with an established resource base in these diseases might compensate. In the long term, fragility should be offset by stability through greater global coordination. Longitudinal and granular data on the geography of disease and research, such as presented in this article, are essential to enable such increased coordination. |

Research has inherent risk and uncertainty[21]; it is difficult to predict which advancements will ultimately lead to widespread benefits[22]. Furthermore, there are often long delays between basic science and its eventual application to health. Therefore, a responsible portfolio of research must balance between reacting to current burdens and anticipating future demands. However, balance at the level of a nation-state does not necessarily translate to global balance. In the 1990s, the Global Forum for Health Research coined the 10/90 gap to emphasize that 10% of health research funding globally was devoted to diseases that encompassed 90% of the global disease burden[23]. Although the precise empirical basis for this statement has been debated, there is consensus that many high-burden diseases receive disproportionately little research attention, which mostly disadvantages low- and middle-income countries[5,24].

While previous studies shed light on the research–disease divergence at specific points in time, there has been little focus on how this divergence has evolved diachronically. Without longitudinal analysis, assessment of progress and possible actions to better align research priorities with societal demands remain limited. Previous studies have mostly relied on the International Classification of Diseases (ICD) system[25] and Medical Subject Header (MeSH) terms to create a larger scale crosswalk between disease burden and disease-related research[3,5]. However, ICD codes are primarily designed for purposes such as healthcare resource allocation, billing and reimbursement, which may also be country-specific[26,27]. ICD codes were not designed to link global disease burden to scientific publication data. Consequently, this approach may only capture a subset of relevant publications, that is, those that unambiguously match disease classifications across data sources, while missing publications that appear ambiguous to this approach and would require an expert contextual assessment on millions of possible publications.

This study addresses the gap in our understanding of how life science research has evolved in relation to changes in the global burden of disease (GBD). We apply a large language model (LLM) in a triangulated approach to create a comprehensive and granular crosswalk between global research and disease burden datasets covering two decades. Our approach is validated against both the ICD-based crosswalk and physician-derived ground truth, demonstrating strong improvements

in accuracy (Extended Data Fig. 1). In addition, we integrate the geographical data of authors and research funders, drawn from both PubMed (https://pubmed.ncbi.nlm.nih.gov/) and Web of Science (https://www.webofscience.com/), including funding information from about four million funding acknowledgements in research articles and more than 50 million geographically designated authorships. Extended Data Fig. 2 summarizes data construction. These enriched data allow us to construct a longitudinal measure of the divergence between life science research and the burden of disease at global and regional levels, analyzing 8.6 million disease-specific research articles (1999–2021) and linking these articles to time-varying data from the GBD database[28]. Table 1 summarizes our main findings and policy implications.

## Results

### Research and burden of disease

To assess the degree of alignment between research and burden of disease, we built on previous studies by relating the distribution of research publications across diseases to the distribution of disease burden (DALY metric). The DALY metric quantifies the total number of years of life lost because of illness, disability or premature death[29]. Alignment was defined by the extent to which the proportion of global research devoted to a particular disease corresponded to the proportion of global DALYs attributed to that disease[3,5].

We first replicated previous cross-sectional results by pooling our longitudinal data of research publications published between 1999 and 2021. We plotted the distribution of disease-specific research publications worldwide (Fig. 1a, blue bars) against the distribution of DALYs associated with these diseases (Fig. 1a, red bars). Divergence between research and disease burden was observed in both directions: research production was relatively greater than the disease burden for disease areas such as neoplasms, neurological disorders and digestive diseases, whereas research production fell short of the relative disease burden for cardiovascular diseases, maternal and neonatal disorders, and respiratory infections (among others). Only in a few disease areas, for example, diabetes and kidney diseases, we observed relatively close alignment.

The divergence in disease-specific research was accompanied by geographical imbalance in research production (Fig. 1b). Countries

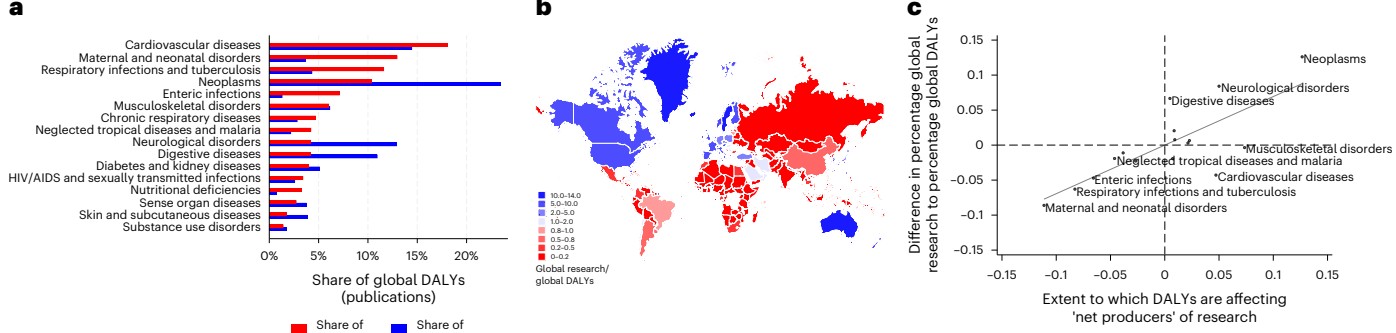

**Fig. 1 | Research versus disease burden stratified according to disease and country. a**, Research versus disease burden stratified according to disease and pooled for the years 1999–2021. The average share of global DALYs for each disease category (red bars) compared to the average share of global research articles on each disease category (blue bars). **b**, Research-to-disease burden ratio according to country, pooled for the years 1999–2021. The country's share of global DALYs aggregated for 16 disease categories relative to the country's share of global research articles. **c**, Correlation between disease burden locality and research-to-disease burden ratio according to disease, pooled for the years 1999–2021. Scatter plot of the difference between each disease's global DALY share and its share of global research (*y* axis) versus the difference in DALY shares between net producer countries and others (*x* axis) (DALY share in blue-colored countries in **b** minus that in red-colored countries). Fig. 1 is based on 8.6 million disease-specific research articles across 16 level-2 disease causes as defined by the GBD database.

in North America, Europe and Oceania contributed a larger share of global research relative to their share of disease burden (blue-shaded countries, 'net producers'), whereas countries in Asia, Africa, Latin America and the Caribbean contributed less research relative to their disease burden (red-shaded countries).

The divergence in disease-specific research (Fig. 1a) and the imbalance in the geography of research production (Fig. 1b) were strongly correlated, as shown in Fig. 1c. When examining disease-specific research produced by North America, Europe and Oceania (the blue-shaded 'net producer' regions in Fig. 1b), a clear pattern emerged: diseases that predominantly affect research-intensive regions were also the ones that tended to be disproportionately researched relative to their burden.

### Longitudinal assessment of divergence

Central to our assessment of divergence was the normative principle that the distribution of research should follow the distribution of disease burden. Therefore, we used the Kullback–Leibler divergence (KLD) as our main metric to capture the degree to which a distribution of research resembles that of a reference distribution (DALYs)[30,31]. In addition to the KLD, we constructed other divergence metrics and obtained consistent results (Supplementary Fig. 1).

Contrasting the stark global divergence observed in the cross-sectional analyses (Fig. 1), the longitudinal assessment shows an almost monotonic decline of divergence between research and disease burden from 1999 to 2019 (Fig. 2a). During this period, the KLD measure of divergence decreased by about 50%. When the two most recent years with available data on DALYs were added (Fig. 2a, dashed line), the reduction in divergence was about 60% relative to the base year of 1999. The marked reduction in divergence in the two years of 2020 and 2021 is attributable to the coronavirus disease 2019 (COVID-19) pandemic (Fig. 2b).

The proportion of DALYs caused by respiratory infections and tuberculosis, the disease category in the GBD data that captures DALYs caused by COVID-19, increased from about 8% (2019) to 13% (2020) and then to a 16% (2021) share of global DALYs. At the same time, research on COVID-19 also increased sharply[32,33]. While 3% of global disease-specific research was related to respiratory infections and tuberculosis in 2019, this figure quadrupled to 12–14% in 2020–2021. In other words, there was a COVID-19-related increase in DALYs of 5–8 percentage points and a COVID-19-related increase in research output of 9–11 percentage points. Thus, the COVID-19 pandemic highlights

the capacity of the global research system to align rapidly with emerging health challenges. However, given the exceptional nature of the COVID-19 pandemic, we continued with a conservative analysis of the research–disease divergence by focusing on the pre-pandemic years (that is, 1999–2019). Our finding of a declining research–disease divergence remained consistent when analyzing prevalence and mortality as separate metrics, instead of the combinatorial DALY metric (Extended Data Fig. 3).

### Contributors to declining divergence

To disentangle whether changes in research or disease burden contributed to the overall declining divergence, we constructed two hypothetical scenarios. In the first scenario, we held the distribution of research constant as observed in 1999, allowing only the distribution of DALYs to vary over time. Fixing research and varying the DALY distribution tests how the divergence would have evolved had the research enterprise not changed since 1999. The near-perfect collinearity between this hypothetical scenario (Fig. 3, green line) and the observed trend (Fig. 3, blue line) shows that changes in the distribution of research across diseases had minimal impact on reducing divergence. Over the 21-year period, the proportion of research publications devoted to specific diseases remained virtually unchanged (Extended Data Fig. 4).

Next, we tested a second scenario, fixing the distribution of disease burden as observed in 1999 and allowing only the distribution of research to vary over time (Fig. 3, red line). The results show that if the distribution of DALYs had not changed over the years, there would have been no reduction in the divergence. From 2006 onward, changes in DALYs were statistically distinct from changes in research, indicating that changes in the GBD, rather than changes in research, led to a reduction in divergence. This scenario-based analysis produced consistent results when analyzing mortality and morbidity, respectively (Supplementary Fig. 2). A geographical stratification of the underlying research and DALY distributions is provided in Supplementary Fig. 3. A summarizing visual of the relative lack of changes in research vis-à-vis DALYs according to disease is depicted in Supplementary Fig. 4.

Given the minimal adjustments in the global research enterprise to disease burden over the past two decades, we examined whether research responds with a time lag to changes in the burden of disease[34,35]. Specifically, we compared the distribution of DALYs in a given year with the 10-year lagged distribution of research and obtained no evidence that research adjusted to the burden of disease, either diachronically or with a lag of up to 10 years (Extended Data Fig. 5).

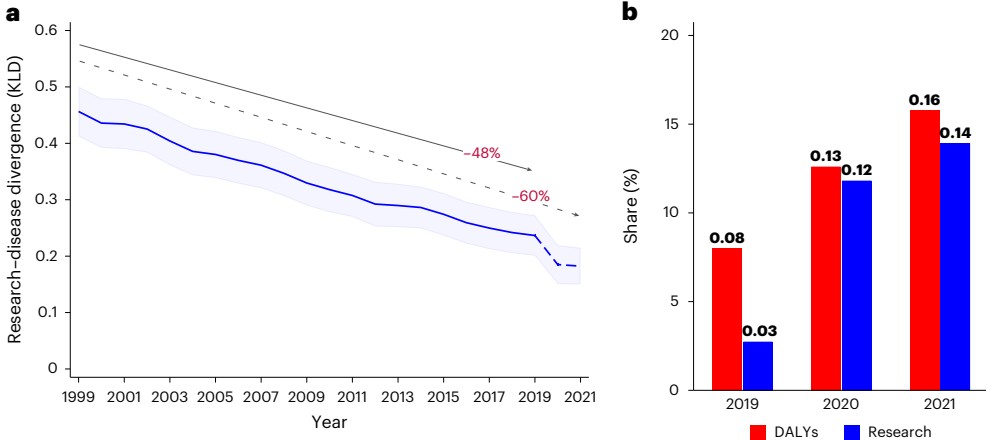

**Fig. 2 | Evolution of research–disease divergence and share of research and disease burden for respiratory infections and tuberculosis. a**, Evolution of research–disease divergence for the years 1999–2021. Research–disease divergence (KLD) of global research share versus global DALY share for each disease category and each year, based on 8.6 million disease-specific research articles across 16 level-2 disease causes. **b**, Share of research and disease burden for respiratory infections and tuberculosis for the years 2019–2021. Share DALYs (red bars) versus share research articles (blue bars) for respiratory infections and tuberculosis as the percentage of global DALYs and global research in the respective years, based on 174,654 research articles focused on respiratory infections and tuberculosis (including COVID-19) for the years 2019–2021.

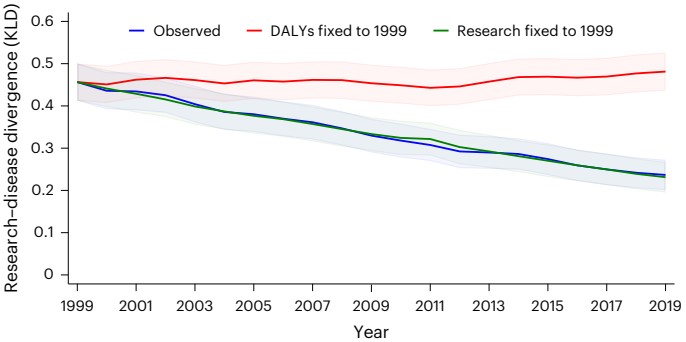

**Fig. 3 | Contributions of research and disease burden to changes in divergence for the years 1999–2019.** Research–disease divergence (KLD) of global research share versus global DALY share for each disease and each year (1999–2019) with observed data (blue line) and simulated distributions assuming constant 1999 research shares (green line), and constant 1999 DALY shares (red line), based on 7.4 million disease-specific articles across 16 level-2 disease causes.

We further examined whether there were differences in certain types of research in adjusting for changes in disease burden. First, we compared basic research to applied and clinical research, finding that these different types of research showed no differences in adjustments (Extended Data Fig. 6). When we compared research publications that acknowledge funding to publications that do not, we also found no differences. Additionally, we identified publications with industry involvement and also observed no differences. Only when we considered industry involvement specific to late-stage clinical trials, we obtained a material difference such that late-stage drug development showed greater divergence from burden of disease, which also declined at a lower rate over time (Supplementary Fig. 5a–c). Overall, across different strata of research, we found little to no evidence that the global research enterprise has contributed tangibly to the observed decline in the research–disease divergence.

## Differentiating diseases in their contribution to changing divergence

As the distribution of disease burden has driven the reduction in the research–disease divergence, we examined the extent and direction in which diseases have contributed. Figure 4 shows two important dichotomies. First, the diseases that reduced divergence are predominantly communicable diseases, while those that increased divergence are exclusively noncommunicable diseases. Second, the diseases that reduced divergence are geographically localized. In contrast, the noncommunicable diseases that increased the divergence between research and disease burden are global. We used the Herfindahl–Hirschman Index (HHI) to quantify the geographical concentration of disease burden (Extended Data Fig. 7). The differentiation of diseases highlights the geographical and disease-specific dynamics underlying the global alignment and misalignment of research and health priorities.

## Geographical variation in research and disease burden

To further examine geographical variation in research–disease divergence, we analyzed the evolution of both DALYs and research output in eight different world regions, as defined by the United Nations[36]. We focused on diseases that contributed to a change in overall divergence of at least five percentage points over the past 20 years (Fig. 4). These include cardiovascular diseases, which are prevalent worldwide, representing global diseases, and respiratory infections and tuberculosis, enteric infections, maternal and neonatal disorders, and nutritional deficiencies, which are more geographically concentrated, representing local diseases.

Extended Data Fig. 8 (left panel) shows the absolute change in the share of DALYs and research for these local and global diseases from 1999 to 2019, aggregated across world regions. Consistent with our composite measure of research–disease divergence (KLD), we found that the share of research devoted to these diseases has essentially not changed. However, the share of DALYs has shifted, with the burden of cardiovascular diseases increasing by approximately 4 percentage points, while the burden for local diseases has declined at three times that rate. Decomposing the effects according to geographical region, we found variation in the adjustment of local research relative to the regional increase in DALYs due to cardiovascular diseases. In Central and Southern Asia, the increase in DALYs exceeded the growth in research by nine percentage points, whereas in the next most-affected regions, East and South-East Asia, North Africa and West Asia, the difference was seven and four percentage points, respectively (Extended Data Fig. 8, top right panel). Conversely, the research-intensive regions of North America, Europe and Oceania have maintained relatively stable research enterprises, while the burden of cardiovascular diseases has declined in these regions.

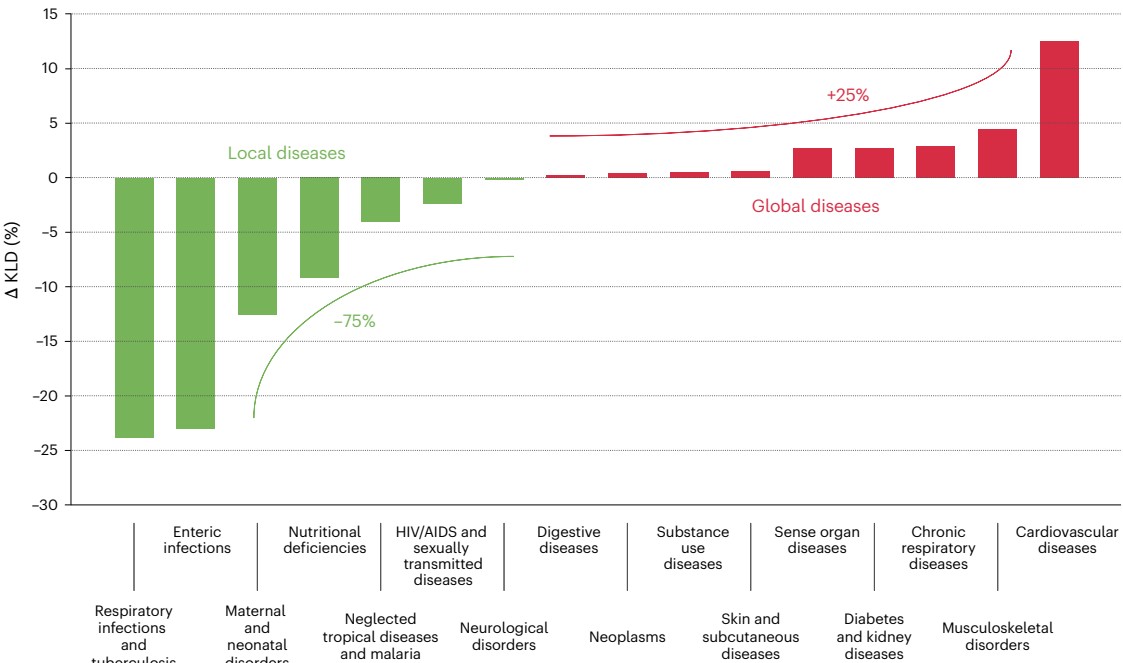

**Fig. 4 | Contributions of specific disease categories to changes in divergence.** Each disease category's contribution to the relative change in research–disease divergence (KLD) from 2019 to 1999. Diseases are categorized according to their geographical concentration of disease burden (HHI) into local diseases (green-colored bars) and global diseases (red-colored bars), based on 708,965 disease-specific articles from the years 1999 and 2019 across 16 level-2 disease causes.

The regional breakdown for the local diseases shows that only sub-Saharan Africa has increased its relative research effort on these four diseases over the past two decades (Extended Data Fig. 8, bottom right panel). We found that sub-Saharan Africa also suffered the highest burden of these diseases (Extended Data Fig. 7), which together accounted for more than half of the region's total disease burden in 2019. In contrast, other regions showed little or no adjustment in their local research efforts, except for Central and South Asia, which also experienced the largest decline in corresponding DALYs. However, this decline in DALYs was not accompanied by a corresponding increase in research on cardiovascular diseases in the region, despite Central and South Asia experiencing the largest rise in DALYs from cardiovascular diseases compared with all other regions of the world.

**Future projections and impact of US public funding withdrawal**
We used the results of our retrospective analyses to shift the focus to the likely future evolution of the global research–disease divergence. We generated two simulations. In the first model (scenario 1), we projected future research trends for each disease based on the trajectories observed at 4-year intervals through 2019. We projected the likely distribution of disease burden for the same set of diseases, building on previous literature assuming a continuation of past progress, which already reflects, to some extent, longer-term societal dynamics, including economic and demographic changes[37] (Fig. 5a, blue dashed line).

In the second model (scenario 2), we considered the likely impact of the changing research and funding landscape under the current US administration. To do so, we subset all research publications that acknowledged receipt of US public funding and stratified the publications according to disease focus and the geographical location of the first author (accounting for the geography of all authors did not change the results; Supplementary Fig. 6). We then simulated how the divergence would evolve in the absence of research from non-US-affiliated first authors who received US public funding. To ensure comparability between scenarios 1 and 2, we again assumed the same trend in the distribution of disease burden (Fig. 5a).

The projection of the research–disease divergence in scenario 1 suggests that the global divergence will, at best, stabilize over the next 10 years. The reductions observed in recent decades, driven by declines in DALYs from communicable diseases, are unlikely to contribute to further reductions in the research–disease divergence at historically observed rates. Instead, under the assumption of linear trends, the simulation points to an increasing divergence between research and disease burden in the future. Across 1,000 independent simulations, the divergence is projected to increase by about 50% (KLD of 0.27 in 2050 relative to KLD of 0.18 in 2021).

By contrast, a withdrawal of US public research funding would lead to an abrupt and large increase in divergence. Compared to scenario 1, we simulated a 25% increase in divergence (KLD of 0.25 versus 0.20) if US public funding for international first authors were to be withdrawn over the next 5 years. A sustained withdrawal of US public funding over the next 20 years would even result in a reversal of almost half of the reduction in divergence observed over the past 20 years. These simulations indicate the fragility of the interconnected global life science research enterprise and the potential consequences for the research–disease divergence in both the short and long term. Supplementary Fig. 7a–c provides the underlying disease-specific projections for research and disease burden.

To identify which areas of global research are most vulnerable to potential cuts in US public funding, we stratified publications receiving US public funding according to disease area and first-author region (Fig. 5b). Sub-Saharan Africa is the most exposed geographical region, with US public funding supporting over 20% of region-led publications in several disease categories. Among the most exposed diseases, 25% of research on respiratory infections and tuberculosis, and 41% on HIV/AIDS and sexually transmitted infections, received US financial support. The exposure to US public funding extends beyond communicable diseases. Research on substance use disorders and neurological disorders also shows high dependency on US public funding. Across regions, US public funding has an important role in sustaining research on both infectious and chronic diseases, underscoring the global reach, and potential disruption, of any retrenchment.

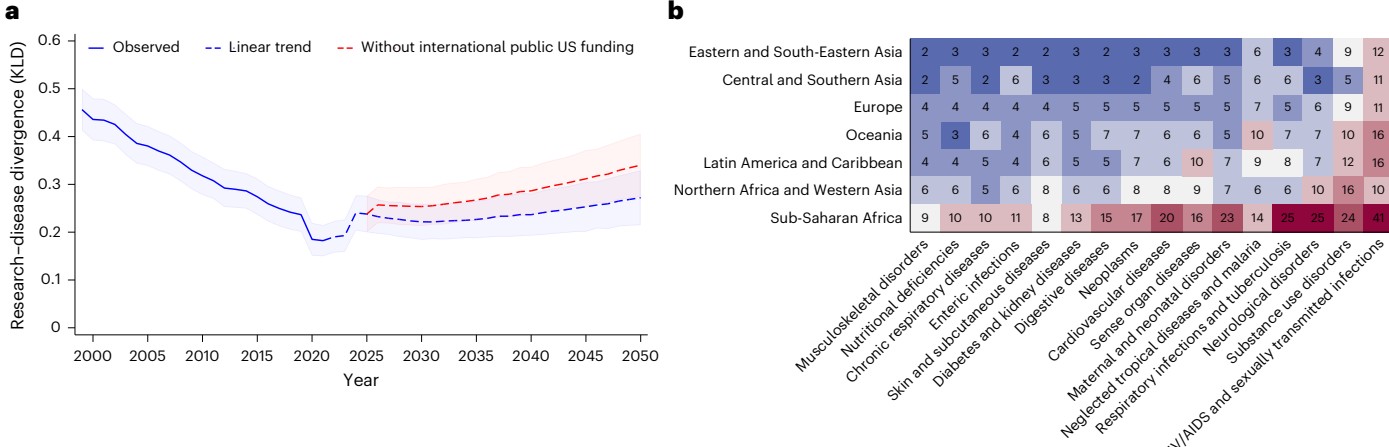

**Fig. 5 | Projected future evolution of research–disease divergence up to the year 2050 and exposure to US public research funding according to geographical region and disease. a**, Simulated research–disease divergence (KLD) under scenarios of linear growth in research output (dashed blue line) and linear growth in research output without the contribution of US public research funding to internationally led research (dashed red line). Both scenarios (that is, dashed blue and red lines) simulate continued historical progress in DALYs. Simulated values are based on 480,000 simulations of DALY estimates and yearly research article counts across 16 level-2 disease causes for 30 years. **b**, Share of research publications that acknowledged receipt of US public research funding, disaggregated according to first-author region (excluding North America) and 16 level-2 disease causes for the years 2015–2021.

To gauge which countries may be positioned to compensate for a decline in US public research funding, we analyzed national funding shares across the most-affected disease areas (Extended Data Fig. 9). Focusing on the five diseases most dependent on US public funding, we calculated, for each disease, the share of articles funded by institutions from the ten non-US countries with the highest number of funding acknowledgments in our dataset. Although general research funding does not equate to a country's preparedness to support science internationally, we found that China accounted for the largest share of non-US funding in both respiratory infections and tuberculosis (20%) and neurological disorders (16%), indicating potential capacity to fill emerging gaps in these domains. The United Kingdom leads in funding HIV/AIDS and sexually transmitted infections research (16%), while Canada (12%) and Australia (10%) are notable funders of research on substance use disorders. These data show the concentration of alternative research and funding capacity in a limited set of countries and highlight the risk of research disruption in low-resource settings should US support recede without international coordination. The findings also signal a growing polarization among research-intensive regions, with the potential for deepening disparities in global health research, if funding from dominant countries diminishes.

## Discussion

This study provides a comprehensive assessment of the alignment between global research and GBD over time. The data show a seemingly increased alignment of research and burden of disease over the past two decades, driven almost entirely by changes in the burden of communicable diseases, while research has remained largely unchanged. These contrasting patterns may be problematic. The social contract underlying the public funding of research suggests that in a world of changing diseases and associated societal needs, research is incentivized to address those needs. We find little evidence of adjustments in research, either at the global or regional level of the research enterprise (except for sub-Saharan Africa), or across different types of research from basic to clinical, or across funding characteristics.

Research resources are finite. The risk of a laissez-faire approach of indiscriminate research proliferation is neither sustainable nor ethically defensible in a world of shifting health challenges. Yet, targeting research more directly towards health impact is complicated by the fundamental uncertainty of what and whether research will ultimately deliver widespread benefits[22]. This uncertainty underscores the importance of a procedurally just approach to distributing research efforts, one that responds to current and future health challenges and promotes alignment between research activity and disease burden over time[38–40].

While research output has remained stable and progress to reduce GBD has been made, we caution against interpreting the current approach as sufficient or low risk. It is plausible that the improved alignment could have been even greater had research responded more directly to evolving health needs, either by influencing disease patterns or by better aligning with them. Our projections indicate that this historical improvement is unlikely to continue. Without intentional redistribution, the alignment between research and disease burden is likely to stagnate, or even decrease, over the next 20 years if the patterns persist.

Past research has advanced two dominant explanations for a stable or inelastic research enterprise. One argument suggests that the adjustment costs of redirecting research are high because scientists are reluctant to change their research programs[35]. The other argument points to latency in the training of the scientific workforce: it takes many years to complete training and reach independence. The resulting subject-specific skills and the need for documented expertise of scientists to secure funding makes it difficult to adjust research programs[41–44], even if scientists might want to. The COVID-19 pandemic proves an exception to both lines of reasoning, with research and funding in many disciplines changing course[32,45]. Funding for COVID-19 was nimble and focused, allowing researchers to respond timely to a societal demand. Policymakers and funders should learn from this, supporting agenda setting through allocating a portion of their funding toward the most pressing local and global (health) issues and providing more efficient mechanisms for the receipt of funding.

Calls to reform the global research enterprise come at a time of profound change in science funding. At the time of writing, the US administration had proposed a 40% cut to the National Institutes of Health budget, from US$ 47 billion down to US$ 27 billion, while terminating existing grants midstream and doubling the rate of grant rejections[46]. Internationally, the US administration has made drastic withdrawals of funding to global health, including for the US Agency for International Development, the President's Emergency Plan for AIDS Relief and the vaccine alliance Gavi, which the United States will

cease to support entirely[47–49]. The fiscal year 2026 budget request eliminates the Centers for Disease Control and Prevention Global Health Center and funding for bilateral programs[50]. A recent study projects that cuts to the US Agency for International Development may result in over 14 million additional deaths by 2030, including more than 4.5 million children under the age of 5 years[51]. Considering the massive cuts across all global health initiatives suggests even starker consequences.

Our projections across the research landscape indicate that the withdrawal of US public funding for non-US-led research will lead to a sharp increase in the divergence of global research and global disease burden, with the potential to reverse two decades of progress over the longer term. Sub-Saharan Africa faces disproportionate risks under this scenario. In 2023, 81% of US global health funding was bilateral and 84% of that was directed to this region[52]. Our analysis shows that 20% of all publications led by sub-Saharan Africa and 41% of those focused on HIV/AIDS and sexually transmitted infections were supported by US funding. The uncertainty caused by abrupt funding terminations affects the entire research enterprise, including international science research networks, capacity-building global health partnerships and multicountry clinical trials[52,53].

The lack of more coordinated international strategies may result in a more polarized research landscape that increasingly diverges from effectively addressing diseases. Broadly, the world is transitioning from infectious-related and poverty-related diseases to chronic noncommunicable diseases, such as cardiovascular diseases, cancer and diabetes, with a high disease burden globally. Paradoxically, as disease patterns become more globally integrated, recent evidence suggests that scientific research is becoming more fragmented, with declining international collaboration overall, particularly in competitive frontier domains, such as artificial intelligence[24,54].

The need for global coordination in health research is well recognized, and past calls for systematic monitoring and evidence-based prioritization remain highly relevant[8]. The current World Health Organization (WHO) Global Health Strategy emphasizes the importance of international collaboration in research and development, alongside strengthened health systems and equitable access to care[55]. However, these calls to action are not always successful. Over 25 years ago, the Global Forum for Health Research was established to advance research on diseases disproportionately affecting low- and middle-income countries[56]. Despite its ambitious mandate, the Forum lacked sufficient financial support to fulfill its intended global role and was ultimately absorbed by the Council on Health Research for Development, which has since also suspended operations because of funding constraints[56]. Beyond underinvestment, past coordination efforts have struggled with institutional fragmentation and competition across countries—challenges that continue to undermine collective action in global health research. This will be particularly difficult with the United States leaving the WHO[57].

This study has several limitations. First, DALYs are subject to critique, particularly regarding the counterfactual construction of their mortality component. Therefore, we complemented them with analyses using mortality and prevalence, which yielded consistent results. To our knowledge, DALYs remain the most comprehensive, globally available and longitudinally consistent metric reflecting both morbidity and mortality, making them well suited for our research question[1,29,38,58]. Second, our analysis focuses on level 2 of the GBD hierarchy to balance interpretability and granularity. While finer resolution at level 3 or subregional scales may reveal additional nuances, our level 3 sensitivity analyses support the main findings (Supplementary Figs. 8 and 9). However, future investigations on a more granular level are warranted. Third, our data on research funding do not fully capture industry contributions. We approximated this through industry-affiliated authorship and late-stage clinical trial analyses, yet dedicated research into private sector activity remains an important future direction,

especially given our call for stronger industry engagement in global coordination efforts.

To close persistent gaps and prevent future divergence between research and disease burden, action is needed across three fronts. First, global research coordination must be recalibrated to reflect the rising dominance of noncommunicable diseases while maintaining vigilance for localized communicable threats. Many diseases currently perceived as 'local' may seem distant from the lived experience in countries with high scientific capacity. The resurgence of measles in the USA illustrates how quickly progress can unravel when public health priorities erode. Similarly, policy shifts, such as the rollback of school nutrition programs in high-income countries, risk reintroducing preventable conditions like nutritional deficiencies, turning once-contained burdens into global concerns and underscoring shared vulnerabilities in a fragmenting system.

Second, durable governance structures are needed to prevent the repeated collapse of coordination mechanisms and to ensure that funding, incentives and accountability are better aligned with global health priorities. Countries should invest science diplomacy efforts because researcher mobility and collaboration are essential to accelerate innovation. These efforts need not be conducted and resourced completely by the public sector. Fragmented national agendas and retreat from global health priorities threaten not only public health but also geopolitical and economic stability, as evidenced by the recent COVID-19 pandemic. Corporations, reliant on a stable and predictable world, have a vested interest in supporting international coordination bodies and research efforts toward global health.

Finally, investment in open science practices and shared data infrastructure, such as the data platform introduced in this study, is essential to enable transparent, real-time tracking of research alignment across geographies and disease areas. Funders should invest in these capabilities and in coordination efforts to ensure timely access to information to inform more evidenced-based approaches to the allocation of resources.

This study underscores the urgent need for a more aligned, equitable and coordinated global health research agenda. Researchers, funders, governments and industry all have a role in strengthening the global science architecture, one that cannot only respond more effectively to today's health challenges but also anticipate those of tomorrow.

## Online content

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

## Methods

### Data sources

**Life science publications.** We extracted life science research publications from the parsed PubMed XML database. Trained librarians associate articles indexed in PubMed with MeSH terms, which constitute a controlled vocabulary for the categorization of biomedical research topics. These terms formed the basis for linking publications to specific diseases in our analysis. Previous studies have also used alternative approaches to designate disease-specific research, including topic modeling[59] or co-word analysis[60]. However, a designation via MeSH terms allows for scalability and consistency across a broad range of diseases[5,6]. We focused specifically on MeSH terms in the 'C-branch' of the MeSH tree, which contains terms related to diseases. As of June 2024, PubMed recorded 5,032 unique C-branch MeSH terms. We started article extraction in 1999, when PubMed began to systematically record first-author affiliations, which we used to geolocate authorships and research articles. The endpoint of our dataset was 2021, the latest year for which comprehensive GBD data were available at the time of writing.

**GBD data.** To assess disease burden, we used data from the GBD database, maintained by the Institute for Health Metrics and Evaluation (IHME)[28]. The GBD database provides detailed, time-varying estimates of DALYs, a metric capturing the cumulative years of life lost due to illness, disability or premature death for specific diseases. DALYs provide a comprehensive metric that enables longitudinally consistent comparisons across diseases and regions, making them uniquely suited for analyzing divergence between research and disease burden over time.

The GBD database is organized according to disease within a hierarchical structure of four levels. Level 1 is the most general in the hierarchy, distinguishing between communicable and noncommunicable diseases. For our main analyses, we focused on level 2, which strikes a balance between granularity and interpretability at the global macro level. This cause level allows us to distinguish, for example, enteric infections as a subcategory of communicable diseases, or cardiovascular diseases as a subcategory of noncommunicable diseases. In line with previous research[3,5], causes that are difficult to assign to a specific disease (for example, 'other noncommunicable diseases') were excluded from the analysis. While we present our main analysis for causes categorized at level 2, we integrated our matching of disease burden and disease-specific research from lower levels in the hierarchy (levels 3 and 4) and rolled the resulting associations up to the described level 2. For example, myocardial infarction (level 4) was rolled up into ischemic heart disease (level 3), which was rolled up into cardiovascular diseases (level 2). In total, we examined 16 level 2 causes of disease.

In addition, we offer a sensitivity analysis using level 3 disease categories, which yielded results consistent with our main findings at level 2, specifically a reduction in divergence between disease burden and research over time driven by changes in DALYs rather than changes in research output (Supplementary Fig. 8). Furthermore, we selected the two major level 2 research areas (cardiovascular diseases and neoplasms) as examples to show how the underlying research and disease burden at level 3 were distributed. The patterns at level 3 closely mirrored those observed at level 2: research on neoplasms remains disproportionately high relative to their burden, while research on major cardiovascular conditions remains underrepresented (Supplementary Fig. 9a,b).

**Linking research with disease burden.** To identify disease-specific research articles, we linked C-branch MeSH terms assigned to PubMed articles with the corresponding disease categories defined by the GBD database. Traditional approaches rely on the ICD system as a crosswalk to bridge these datasets. While probably precise, this method has notable limitations, including structural mismatches between MeSH terms and ICD codes, as well as inefficiencies introduced by the intermediate step of linking ICD codes to GBD categories. Each of the three classification systems—MeSH, ICD and GBD—was designed for different purposes, begetting variance in nomenclature. Crosswalking these three coding systems requires expert judgment by physicians, preventing large-scale linkage of millions of research articles to specific disease burden causes. These shortcomings are particularly problematic for longitudinal analyses that require accurate and consistent associations of research articles with burden data and for studies of geographical regions or research areas with lower research output, where even small numbers of overlooked articles can substantially affect the analyses.

To address these issues, we used a triangulated strategy that combined manual data curation by physicians, an ICD-based approach and an LLM-based methodology. In the first step, we manually curated dataset to serve as a gold standard for validation for cardiovascular diseases, the disease category causing the greatest burden globally and exhibiting the most complex MeSH-Cause nomenclature matrix. Two co-authors who are practicing cardiologists independently reviewed C-branch MeSH terms related to cardiovascular diseases (MeSH branch C14) and matched them to the GBD level 2 cause 'Cardiovascular Diseases' and its subcategories. This manual process ensured high accuracy in matches and resolved residual ambiguities through iterative discussions. Second, we applied the traditional ICD-based approach, mapping MeSH terms to ICD Tenth Revision codes and subsequently linking these codes to the 16 level 2 GBD causes. This process relied on established crosswalks in the Unified Medical Language System. Third, we developed an LLM-based method using ChatGPT (model GPT-4o), directly assessing whether a MeSH term aligned with a specific GBD cause. Extended Data Fig. 1 provides an overview of our approach.

We designed a custom prompt that directs the model to evaluate each of roughly 1 million possible combinations of 5,032 MeSH terms and 180 GBD causes (Supplementary Fig. 10). This method circumvents the need for an intermediate step that first links MeSH to ICD and then ICD to disease cause in a one-to-many MeSH to ICD and many-to-one ICD to cause matching structure. Instead, the LLM approach allows for the simultaneous evaluation of a many-to-many MeSH-to-cause matching structure at scale, including multiple assignments where appropriate.

Comparing the performance of the LLM-based approach to the ICD-based method using the physician-derived gold standard, we observed a substantial improvement in overall accuracy at the MeSH term level—from 50.5% with the ICD-based approach to 86.1% with the LLM approach. At the article level, the LLM approach even achieved an accuracy of 94.9%, compared to 67.0% for the ICD-based method (Supplementary Fig. 11). The greater accuracy at the article level is expected because more frequent MeSH-to-cause linkages occur in more productive research areas with relatively more articles.

The substantial improvement in the identification of disease-specific research using the LLM instead of ICD codes was mainly due to increased recall, where the LLM performed markedly better. Precision was comparable between the two methods, indicating that neither approach was prone to false positives. However, high and stable recall is critical for our longitudinal assessment of divergence because it minimizes the risk of underrepresenting research activities related to specific diseases.

To ensure the robustness of our approach beyond the cardiovascular disease category, we also evaluated the recall performance for other level 2 causes. This analysis assessed whether articles published in disease-specific journals and assigned a C-branch MeSH term were correctly classified as related to these diseases. The LLM-based approach achieved a recall rate of 94.76% compared to 72.71% for the ICD-based method, with little variation between disease causes (Supplementary Fig. 12). These results further validate the LLM-based approach in connecting research to GBD causes.

Overall, the comprehensive evaluation of different methods for identifying disease-specific research highlights the potential of LLMs to improve the linkage between research articles and diseases. The

feasibility of establishing an accurate and reliable link between disease burden and research is also fundamental to monitoring future progress and we share our publication-level data in an online repository[61] (see the 'Data availability' statement).

**Creation of final sample.** The LLM-based association of disease-specific research articles indexed in PubMed with causes of disease in the GBD database resulted in 7.5 million unique articles published between 1999 and 2021. As a single article can be associated with multiple causes of disease, this resulted in 9.7 million disease-cause article links. We assigned these publications to geographical regions based on the affiliation information associated with the first authors of research articles. In the life sciences, authorship norms associate first and last authors with leading roles in the research project. A comparison of the geographical locations of first and last authors showed that in over 90% of first–last author combinations, the geographical locations were identical at the country level, which is the most granular level of analysis in our study. We used the affiliation information recorded in PubMed and supplemented this with affiliation information from Web of Science. As the affiliation data were recorded as unstructured text, we used ChatGPT to process and assign countries to the affiliation strings, considerably enhancing the curated dataset. We also conducted a separate analysis based on the presence or absence of industry-affiliated co-authors, using information extracted from the authors' institutional affiliations. We randomly selected 200 samples and had two independent raters assess the accuracy of the LLM-assigned country designations. In all cases (100%), both raters confirmed that the LLM country assignments were correct.

Overall, we geolocated over 25 million unique affiliation strings, including those of non-first authors, and successfully assigned geographical information to 6.7 million unique articles. This corresponds to 8.6 million article-to-cause links, covering approximately 89% of the articles in our sample. While our main analysis focused on first authors, a sensitivity analysis that considered all authors' countries yielded consistent results. Additionally, for a subset of 71 from 100 randomly selected articles where study location could be inferred from the abstract, the first author's country matched the research location in over 85% of cases.

To further enrich our dataset, we integrated article-specific funding information from Web of Science, available since 2008. Using the LLM, we assigned countries to funding agencies based on the acknowledgement text, automating what would otherwise have required extensive manual inspection. We also used this approach to identify major US public funding institutions, defined as those acknowledged in at least 1,000 research publications in our dataset and being financed by US taxpayer money. This process identified funding information for about 40% of disease-specific articles in our sample. It is important to note that while these funding data provide valuable insights into the sources of acknowledged funding, we are cautious about drawing conclusions for publications lacking such information. To our knowledge, these are the most comprehensive funding data currently available[62].

Our final sample consisted of 8.6 million publication–cause links, geolocated based on first-author affiliations. For our analysis, we aggregated these data in several ways and supplemented them with DALY data, which provide year-specific, cause-specific and country-specific assessments of the burden of disease. For our primary analyses, we aggregated country-level data into eight geographical regions defined by the United Nations[36]: Central and Southern Asia; Eastern and South-Eastern Asia; Europe; Latin America and the Caribbean; Northern Africa and Western Asia; North America; Oceania; and sub-Saharan Africa. The sample creation process is summarized in Extended Data Fig. 2.

**Analyses.** To quantify the divergence between research and disease burden we used the KLD. Formally, the KLD is a nonsymmetric measure of the difference between two probability distributions $p(x)$ and $q(x)$ and is given by:

$$\text{KLD}(p(x)\|q(x)) = \sum_{x \in X} p(x) \ln\left(\frac{p(x)}{q(x)}\right)$$

where $p(x)$ represents the reference distribution, in our case the discrete distribution of DALYs per disease $x$ (in percentage), and $q(x)$ represents the discrete distribution of research articles per disease $x$ (in percentage). The sum of the individual divergences of research articles to DALYs per disease $x$ across all 16 diseases in the set $X$ of level 2 diseases from the GBD database yields the KLD as our key divergence measure. The KLD is nonnegative, provides an internally consistent measure of divergence over time and decreases as the fit between the two distributions improves; a KLD of zero would indicate a perfect alignment between research and disease burden. To address the KLD's sensitivity to very small probabilities, we conducted sensitivity analyses excluding near-zero outliers (values below 0.01) from both the numerator and denominator and applied Laplace smoothing. These adjustments yielded consistent results. We provide additional metrics, that is, the Population Stability Index[63], the Hellinger distance[64] and the Jensen–Shannon divergence in Supplementary Fig. 1, obtaining consistent results.

To account for the inherent uncertainty in DALY estimates, we used the asymmetric upper and lower bounds reported by the IHME. For each year, we simulated DALY values from a log-normal distribution parameterized according to the reported mean, upper and lower bounds. This approach captures the nonnegative, skewed nature of DALYs while aligning the simulated distribution with IHME uncertainty intervals. We then calculated the divergence metrics for each simulated draw, averaged these metrics across simulations and used the standard deviation to construct the 95% confidence intervals. Thus, we effectively bootstrapped DALY estimates from their distribution and used these bootstrapped values to estimate the divergence metrics.

**Alternative assessment of disease burden.** DALYs represent a composite measure of morbidity and mortality. To test whether the research–disease divergence varied with our measure of disease burden, we reconstructed the KLD with measures for deaths and prevalence, also provided by the GBD (Extended Data Fig. 3). We observed that the trends in research–disease divergence were broadly parallel across all three metrics. Additionally, Supplementary Fig. 2a,b shows that for each disease burden measure, the divergence would not have decreased in the hypothetical scenario where the disease burden had remained unchanged, which is consistent with the findings based on DALYs. A corresponding breakdown according to disease category is provided in Extended Data Fig. 4.

**Alternative research response times.** To check for the possibility of delayed adjustments of research to changes in disease burden, we ran a time-lagged sensitivity analysis of the declining research–disease divergence. Specifically, we compared the distribution of DALYs each year with the distribution of research 10 years later. This lagged analysis yielded no evidence that research adjusted to changes in the burden of disease, either contemporaneously or with a lag of up to 10 years (Extended Data Fig. 5).

**Alternative assessment of research output.** To capture conceivable variation in research

output based on its potential for human application, we built on the work of Hutchins and colleagues, who developed the Approximate Potential to Translate (APT) metric[22]. This metric is derived from a machine learning model that predicts the likelihood that a given publication will be cited in a clinical trial. A higher APT score indicates a higher probability of eventual clinical citation. In addition, we used PubMed publication types, as defined by the iCite classification[65], to identify clinical research. By combining the APT score with the iCite definition, we categorized publications into three different groups: (1) basic research: APT score lower than 0.5 and not classified as clinical; (2) applied research: APT score of 0.5 or higher and not classified as clinical; (3) clinical research according to iCite. For each of these groups, we calculated the relative proportion of articles devoted to each disease and compared this distribution to the corresponding DALY distribution. This allowed us to assess how closely the distribution of each type of research matched the distribution of disease burden (Extended Data Fig. 6).

To gain more specific insight into clinical trials sponsored by industry, we also created a subset of publications linked to trials registered with ClinicalTrials.gov. This link enabled the identification of trial-related publications and extraction of key metadata, most notably, the sponsor and trial phase, thus facilitating a more targeted assessment of industry-driven research. We focused on industry-sponsored phase 3 trials and found a considerable increase in the research–disease divergence for this subset of research. This finding supports our recommendation for greater industry involvement in the coordination of the research enterprise. We also included an analysis of publications based on the presence of industry-affiliated co-authors, identified through authors' institutional affiliations. For this subset, the results were consistent with our main analyses, suggesting that the greater divergence observed for industry-sponsored phase 3 trials is not solely attributable to industry involvement.

In addition to differentiating research according to its clinical potential and industry involvement, we also examined the role of funding acknowledgements in articles published in 2008 or later. We created two additional subsets of articles: those that explicitly acknowledged funding and those that did not. We then compared the distribution of articles in each subset across diseases to the DALY distribution (Supplementary Fig. 5).

**Geographical stratification.** To assess the geographical distribution of diseases, we first calculated the share of DALYs per world region for each level 2 disease cause across the eight world regions. Extended Data Fig. 7 (left) presents the HHI, a widely used measure of concentration. Higher scores indicate a more regionally concentrated disease. The green shading of the bars (left) corresponds to the green shading in Fig. 4, indicating diseases that contributed to reducing the divergence. These diseases are locally concentrated, as their HHI exceeds the average HHI in our sample, and they are mostly communicable diseases. In contrast, the red-shaded diseases, which are noncommunicable, tend to be more globally distributed. The panel on the right ranks diseases in descending order based on their contribution to reducing the research–disease burden divergence over the past 20 years, with the respective disease burden stratified according to world region. The data show that a set of communicable diseases, concentrated in sub-Sharan Africa and Central and Southern Asia have most contributed to the reduction in research–disease divergence.

To test the sensitivity of these geographical findings to the way in which we geolocate research, we also considered the affiliations of all authors, that is, first, last and middle. The findings remained consistent as the overall distribution of authorship according to region changed only marginally when using any authorship position instead of first authorship alone. However, the data suggest that non-first authorship has a more prominent role in certain regions, particularly for East and Southeast Asia (Supplementary Fig. 6).

**Projections of future divergence.** To estimate how the divergence between research activity and disease burden may evolve in the coming years, we projected future research trends for each disease based on the trajectories observed over the last 4 years in our sample period (2018–2021). For respiratory diseases and tuberculosis, we adjusted these projections to account for the impact of COVID-19 in 2022, 2023 and 2024. In addition, we modeled a second scenario that assumes major US public funding agencies will cease funding research led by non-US first authors. This scenario reflects ongoing dynamics within the US science funding landscape and may be viewed as conservative. The distribution of DALYs was taken from a forecast by Vollset et al.[37], assuming a continuation of progress as was observed in the last years. We provide the disease-specific projections in Supplementary Fig. 7a–c.

### Reporting summary

Further information on research design is available in the Nature Portfolio Reporting Summary linked to this article.

## Data availability

The data assembled for this study are available on figshare and can be accessed at https://doi.org/10.6084/m9.figshare.29401205 (ref. 61). Source data are provided with this paper.

## Code availability

The computer code used to perform the analyses in this study is available on figshare and can be accessed via the following link: https://doi.org/10.6084/m9.figshare.29401205 (ref. 61).

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

## Acknowledgements

We thank L. Miao for her valuable assistance in assembling and interpreting the funding data. We also thank L. Mathez, N. Baumann, T.-H. Vu and O. Slobodenyuk for their excellent research support. M.J.L. and T.W.B. received financial support through the Helmholtz Information and Data Science School for Health (HIDSS4Health). L.S. received financial support through the Dr. Hans Riegel Foundation.

## Author contributions

M.J.L. and C.R.S. devised the original idea. L.S and M.J.L. conceptualized the study. M.J.L. led the study and analyses. L.S., M.B., C.L. and M.J.L. assembled and analyzed the data. L.S., C.L., C.R.S. and M.J.L. drafted and wrote the final manuscript. C.R.S. and T.W.B.

provided critical access to data sources and feedback on methods, results and interpretations. All authors edited the manuscript.

## Competing interests

M.J.L. is a co-founder and shareholder of AaviGen, a cardiovascular gene therapy company. The present study is not related to the company. The other authors declare no competing interests.

## Additional information

**Extended data** is available for this paper at https://doi.org/10.1038/s41591-025-03923-0.

**Correspondence and requests for materials** should be addressed to

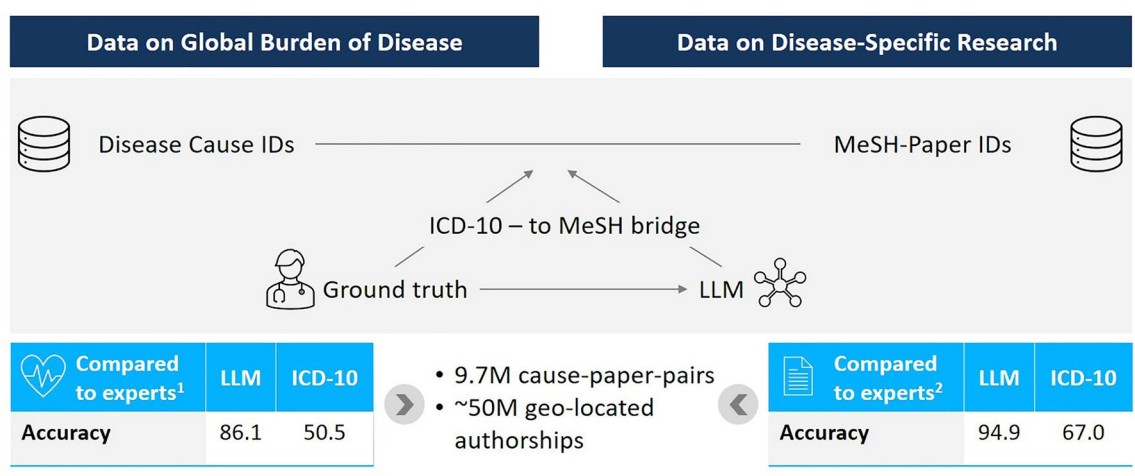

**Extended Data Fig. 1 | Linking global burden of disease causes to research publications.** Schematic visual showing triangulated method for linking disease causes to disease-specific research publications, including a medical expert-derived ground truth, a crosswalk via International Classification of Disease Codes (ICD), and a large language model (LLM) approach using ChatGPT (Model gpt-4o). Accuracy statistics are provided at the MeSH-term level for cardiovascular diseases (467 C-branch terms) as well as at the publication level (1.6 million articles).

Marc J. Lerchenmueller.

**Peer review information** *Nature Medicine* thanks
Fares Alahdab, Md. Mijan Rahman and the other,
anonymous, reviewer(s) for their contribution the peer
review of this work. Peer reviewer reports are available.
Primary Handling Editor: Ming Yang, in collaboration with the
*Nature Medicine* team.

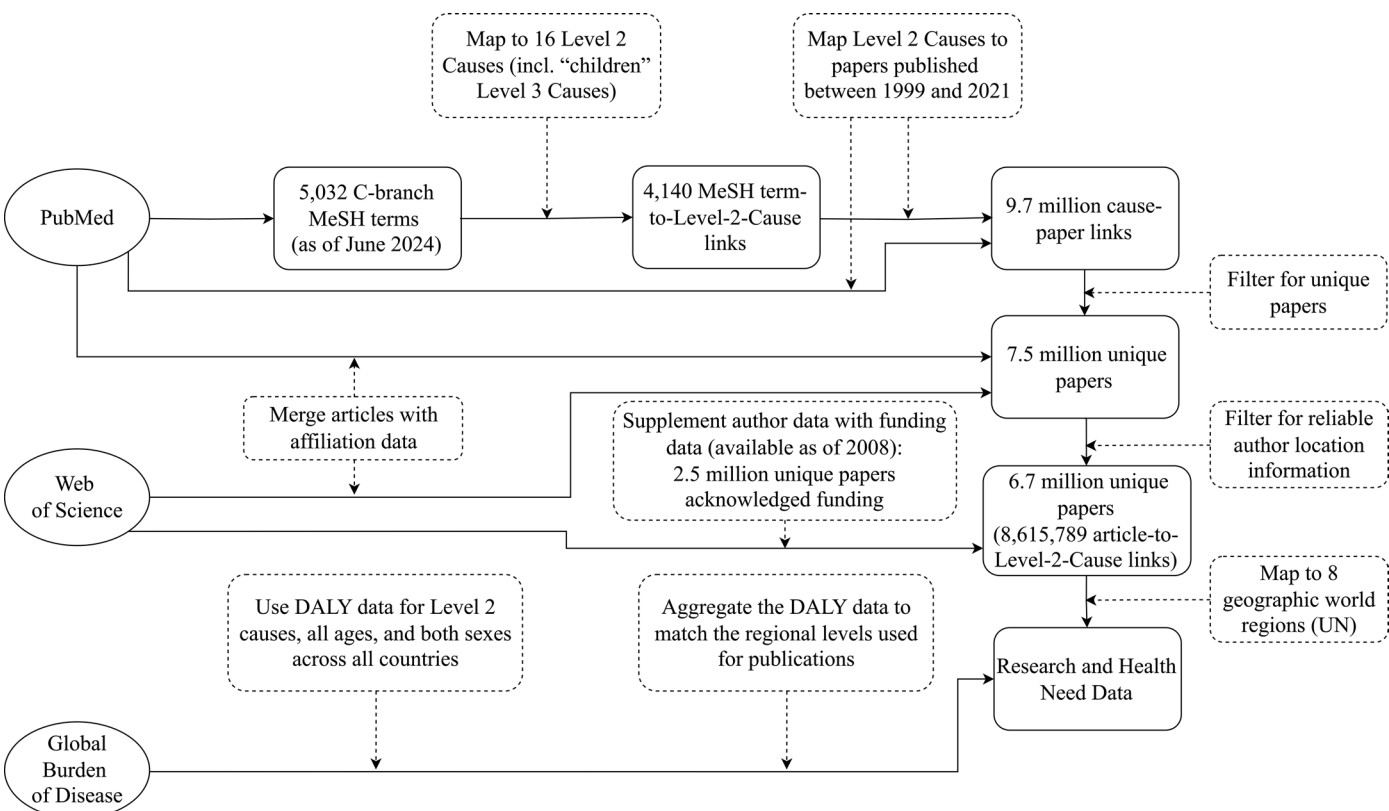

**Extended Data Fig. 2 | Data construction diagram.** Construction of the analytical dataset by integrating publication data from PubMed and Web of Science with disease burden metrics from the Global Burden of Disease (GBD) database.

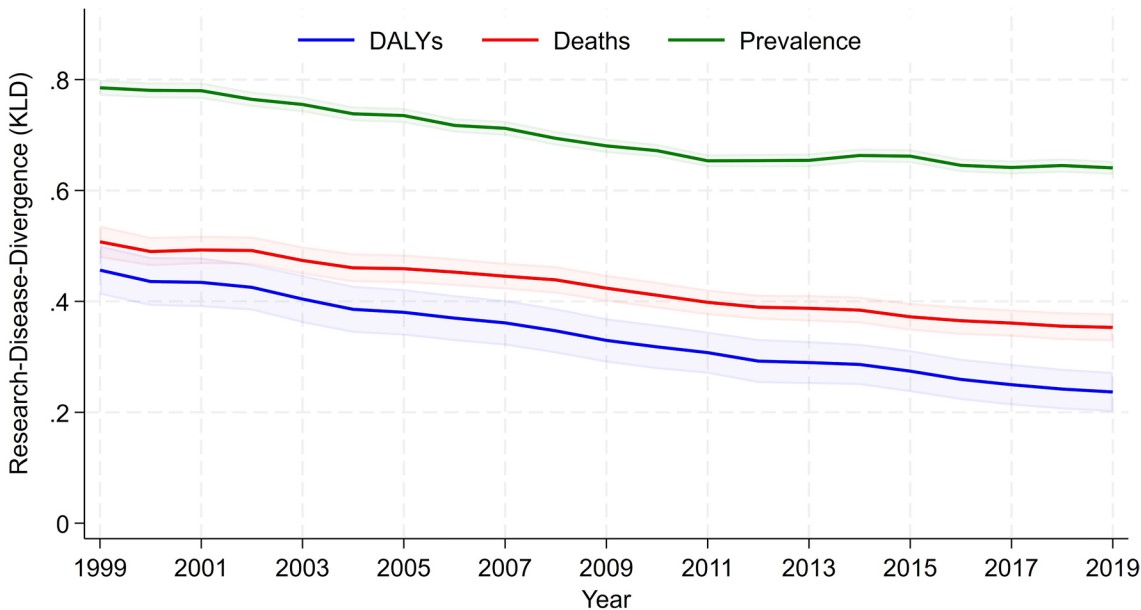

**Extended Data Fig. 3 | Divergence between research and disease burden across different burden measures.** Research-disease divergence (KLD) for different measures of disease burden, including Disability Adjusted Life Years (DALYs, blue line), morbidity (deaths, red line), and mortality (prevalence, green line). Based on 7.4 million disease-specific articles across 16 level 2 disease causes from the years 1999-2019.

## **A.** Proportion of DALYs Caused by Specific Diseases (1999-2019)

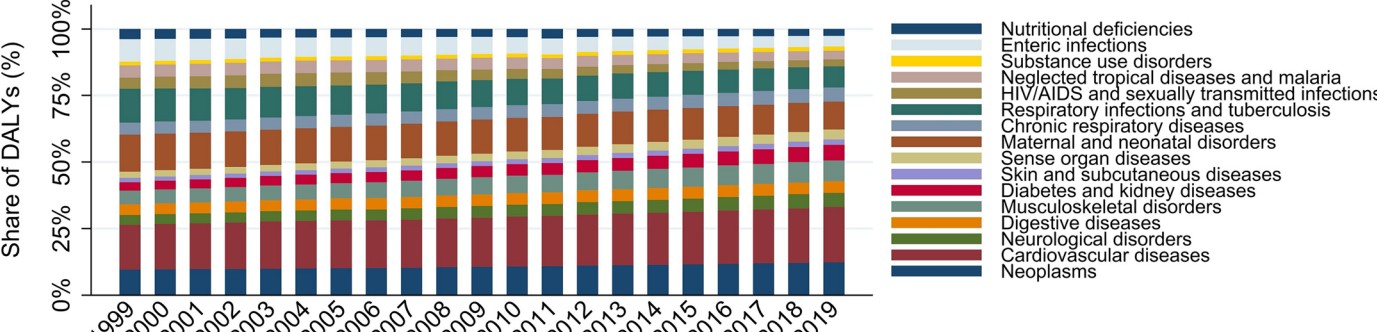

## **B.** Proportion of Research Dedicated to Specific Diseases (1999-2019)

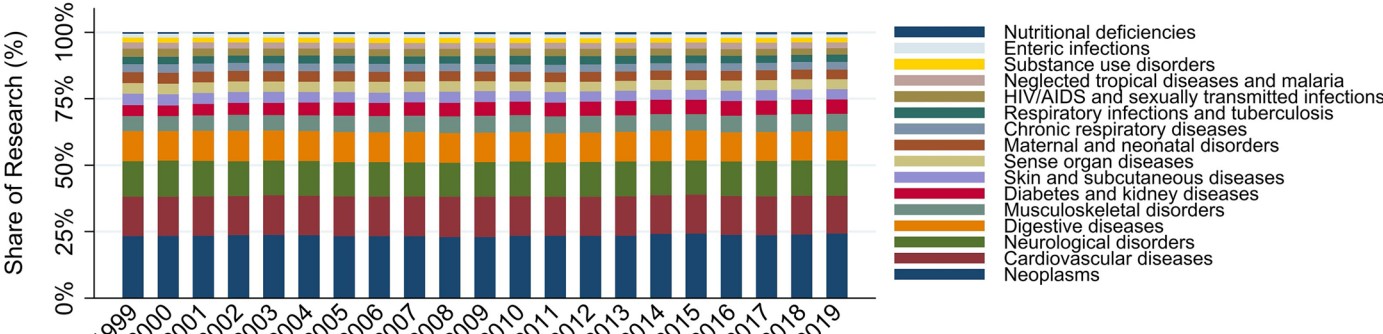

**Extended Data Fig. 4 | Distribution of DALYs (a) and research (b) by disease and year.** Share of Disability Adjusted Life Years (DALYs, panel **a**) by disease category as a percent of global disease burden and share of research articles by disease category as a percent of global disease-related research (panel **b**).

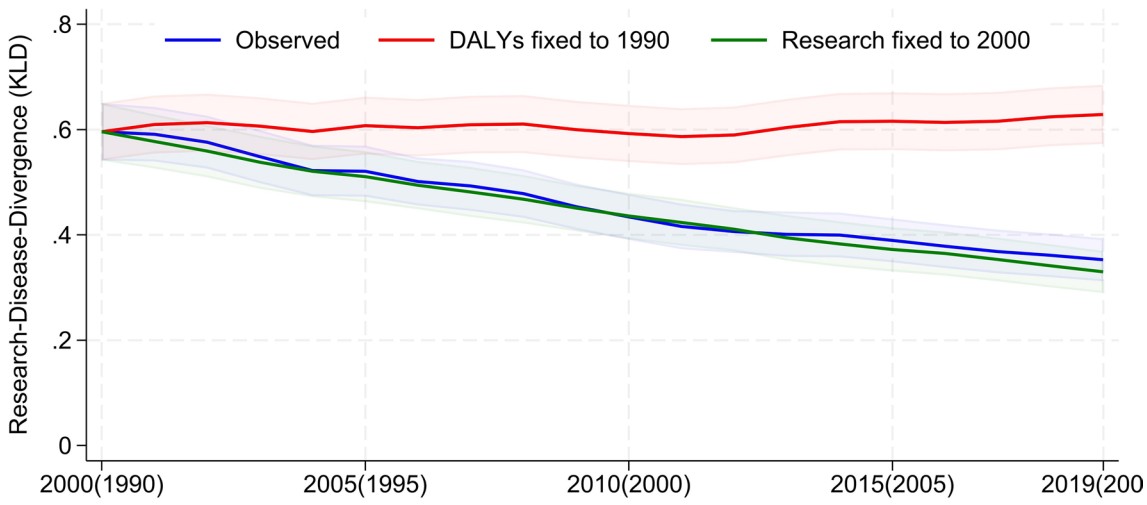

**Extended Data Fig. 5 | Divergence between research and disease burden with a ten-year time lag for research.** Time-lagged sensitivity analysis of the declining research-disease divergence (KLD), comparing the distribution of Disability Adjusted Life Years (DALYs) each year with the distribution of research 10 years later (blue line). The first simulation (green line) fixes research while varying the DALY distribution, testing the thought experiment of how the divergence would have evolved, had the research enterprise not changed since 2000. The second simulation (red line) fixes the DALY distribution to 1990 and considers a time-varying research distribution with a 10-year time lag.

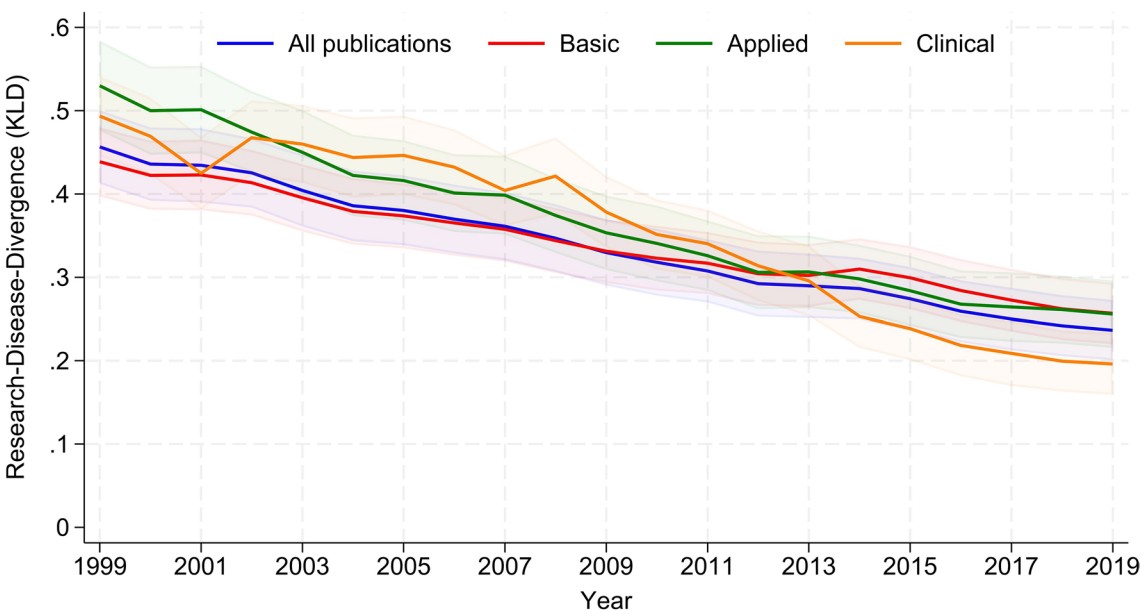

**Extended Data Fig. 6 | Divergence between research and disease burden across basic, applied, and clinical research.** Research-disease divergence (KLD) for different types of research, including basic science, applied science, and clinical studies.

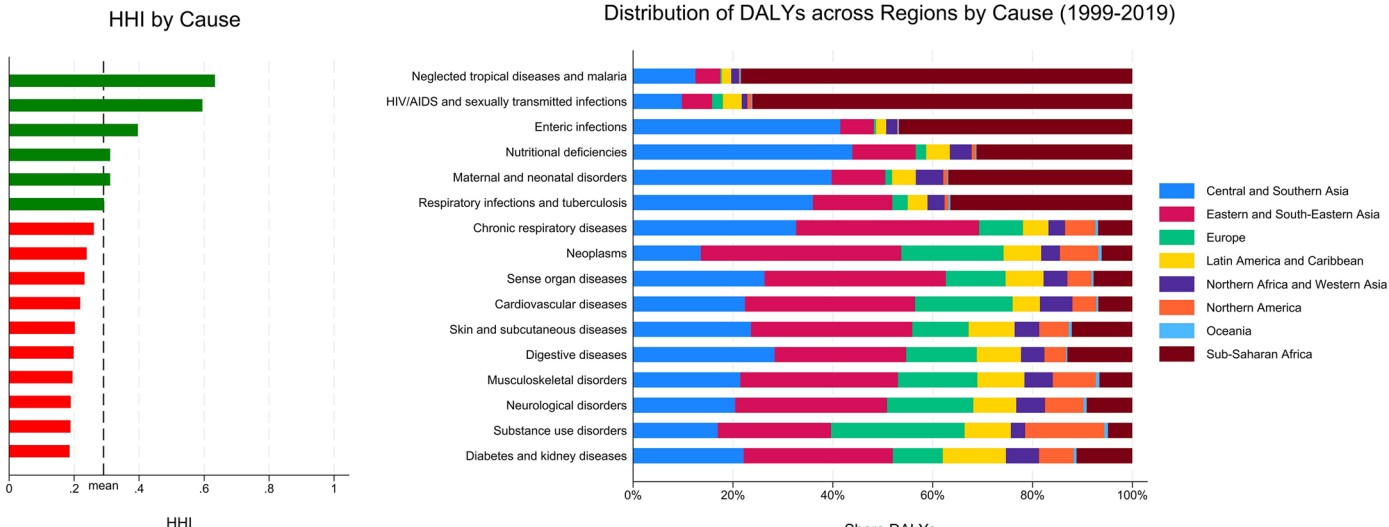

**Extended Data Fig. 7 | Geographic concentration of disease burden across diseases.** The left panel shows Herfindahl-Hirschman Indices (HHI) capturing the regional concentration of disease burden by disease category. Higher HHI values indicate greater geographic concentration. For example, neglected tropical diseases and malaria have the highest concentration (HHI > 0.6), followed by HIV/AIDS and sexually transmitted diseases (HHI ≈ 0.6). Diseases shown in green exceed the global average concentration (indicated by the vertical dashed line) and are classified as 'local' diseases, while those in red fall below the average and are classified as 'global' diseases. The right panel decomposes these HHI values by geographic region. For instance, nearly 80% of the burden from neglected tropical diseases and malaria is concentrated in sub-Saharan Africa.

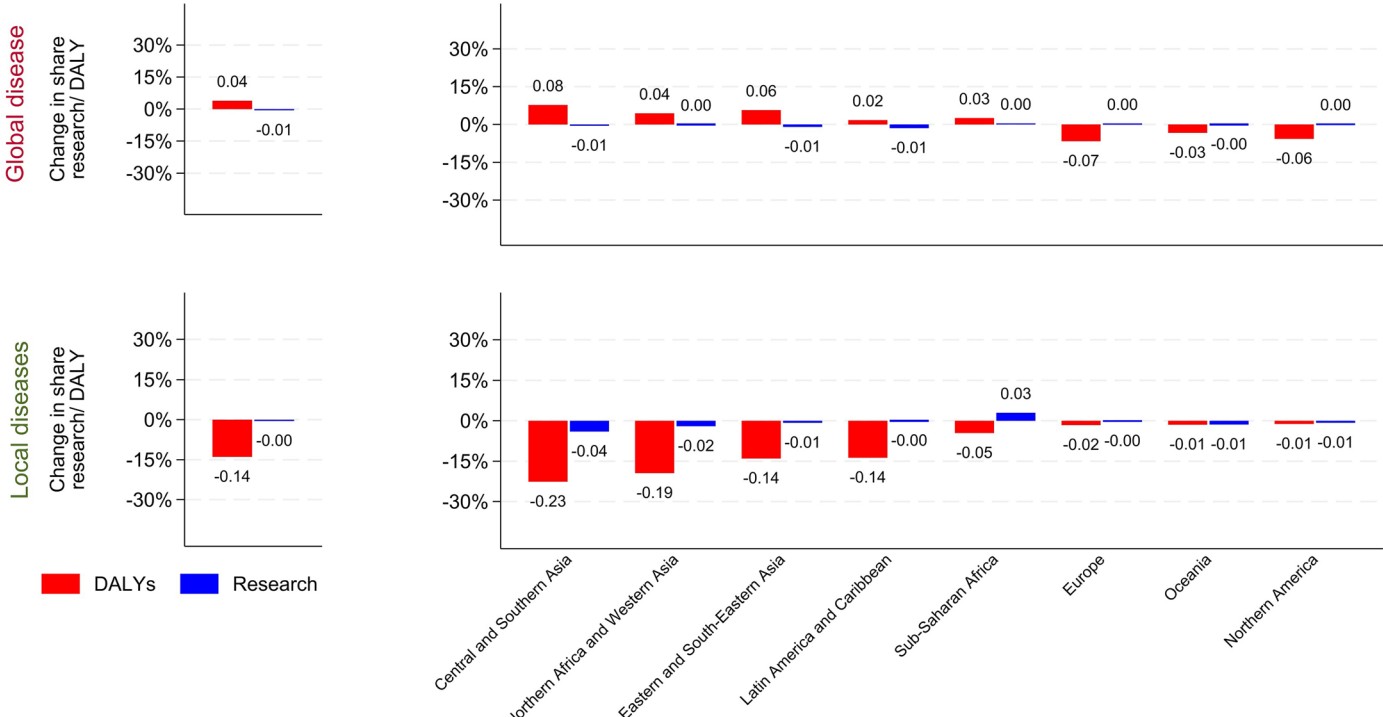

**Extended Data Fig. 8 | Changes in research and burden of disease globally and by region.** Geographic stratification of the five diseases with the largest impact on divergence trends. The top panels highlight a 'global' disease – cardiovascular diseases – that contributes to increasing divergence, while the bottom panels present four 'local' diseases – respiratory infections and tuberculosis, enteric infections, maternal and neonatal disorders, and nutritional deficiencies – that contribute to decreasing divergence. Left panels show absolute changes in global research activity (blue bars) and disease burden (red bars); right panels show the same changes disaggregated by region.

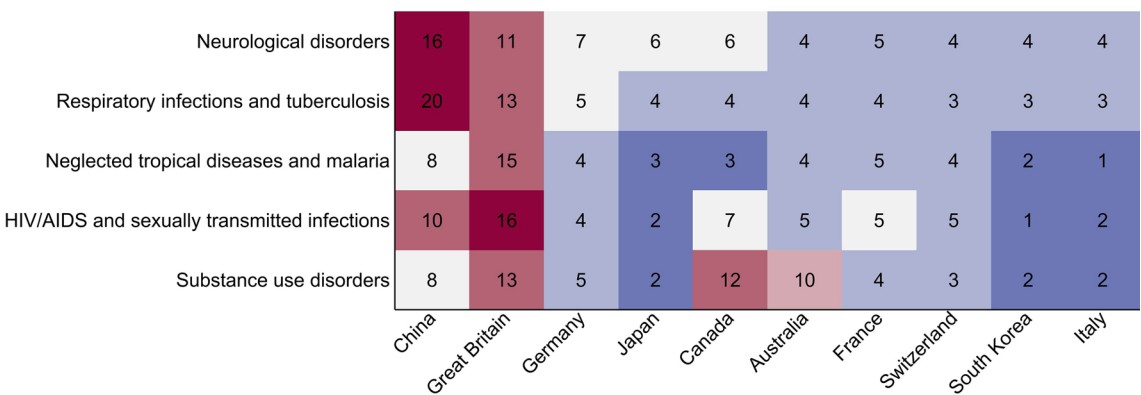

**Share of funded publications**
by top 10 funder country and disease cause - 2015–2021

Darker red = higher share (%); excluding funders from USA.

**Extended Data Fig. 9 | Geography of non-US funders supporting research on diseases primarily backed by major US public institutions.** Share of disease-related articles with acknowledged funding from institutions in the ten leading non-US countries, focusing on the five diseases most dependent on US public support.

# Reporting Summary

## Statistics

For all statistical analyses, confirm that the following items are present in the figure legend, table legend, main text, or Methods section.

| n/a | Confirmed | |
|---|---|---|
| ☐ | ☒ | The exact sample size (*n*) for each experimental group/condition, given as a discrete number and unit of measurement |
| ☒ | ☐ | A statement on whether measurements were taken from distinct samples or whether the same sample was measured repeatedly |
| ☒ | ☐ | The statistical test(s) used AND whether they are one- or two-sided *Only common tests should be described solely by name; describe more complex techniques in the Methods section.* |
| ☐ | ☒ | A description of all covariates tested |
| ☒ | ☐ | A description of any assumptions or corrections, such as tests of normality and adjustment for multiple comparisons |
| ☐ | ☒ | A full description of the statistical parameters including central tendency (e.g. means) or other basic estimates (e.g. regression coefficient) AND variation (e.g. standard deviation) or associated estimates of uncertainty (e.g. confidence intervals) |
| ☒ | ☐ | For null hypothesis testing, the test statistic (e.g. *F*, *t*, *r*) with confidence intervals, effect sizes, degrees of freedom and *P* value noted *Give P values as exact values whenever suitable.* |
| ☒ | ☐ | For Bayesian analysis, information on the choice of priors and Markov chain Monte Carlo settings |
| ☒ | ☐ | For hierarchical and complex designs, identification of the appropriate level for tests and full reporting of outcomes |
| ☒ | ☐ | Estimates of effect sizes (e.g. Cohen's *d*, Pearson's *r*), indicating how they were calculated |

*Our web collection on statistics for biologists contains articles on many of the points above.*

## Software and code

Policy information about availability of computer code

| Data collection | No software was used to collect data. |
|---|---|
| Data analysis | All analyses were performed in Stata 18. |

For manuscripts utilizing custom algorithms or software that are central to the research but not yet described in published literature, software must be made available to editors and reviewers. We strongly encourage code deposition in a community repository (e.g. GitHub). See the Nature Portfolio guidelines for submitting code & software for further information.

## Data

Policy information about availability of data

All manuscripts must include a data availability statement. This statement should provide the following information, where applicable:
- Accession codes, unique identifiers, or web links for publicly available datasets
- A description of any restrictions on data availability
- For clinical datasets or third party data, please ensure that the statement adheres to our policy

We collected data from PubMed via the PubMed XML bulk download, Web of Science (WoS), iCite from the NIH Collection, and the Global Burden of Disease database from healthdata.org.The data assembled in this study will be deposited in the Figshare database.

# Research involving human participants, their data, or biological material

Policy information about studies with [human participants or human data](). See also policy information about [sex, gender (identity/presentation), and sexual orientation]() and [race, ethnicity and racism]().

| | |
|---|---|
| Reporting on sex and gender | N/A |
| Reporting on race, ethnicity, or other socially relevant groupings | N/A |
| Population characteristics | N/A |
| Recruitment | N/A |
| Ethics oversight | N/A |

Note that full information on the approval of the study protocol must also be provided in the manuscript.

# Field-specific reporting

Please select the one below that is the best fit for your research. If you are not sure, read the appropriate sections before making your selection.

☐ Life sciences   ☒ Behavioural & social sciences   ☐ Ecological, evolutionary & environmental sciences

For a reference copy of the document with all sections, see [nature.com/documents/nr-reporting-summary-flat.pdf]()

# Behavioural & social sciences study design

All studies must disclose on these points even when the disclosure is negative.

| | |
|---|---|
| Study description | This study uses a quantitative design to develop a novel dataset linking global disease burden data with disease-related scientific publications. This enables a longitudinal analysis of the alignment between health needs and research activity. The resulting data foundation supports evidence-based agenda setting across different levels of disease specificity and geographic context. |
| Research sample | This study draws on existing datasets from two primary sources: (1) the Global Burden of Disease (GBD) database, which provides level 2 disease categories and corresponding Disability-Adjusted Life Years (DALYs) for all ages and both sexes from 1999 to 2021 across 204 countries and territories, and (2) PubMed, which indexes biomedical research articles annotated with Medical Subject Headings (MeSH). Using a large language model (LLM)-based approach, we linked MeSH terms to GBD disease categories, resulting in 9.7 million article–disease cause connections. The sample includes the global scientific literature and population-level disease burden data, making it representative at the country and disease levels. This approach enables a comprehensive assessment of how research outputs align with health needs over time and across regions. |
| Sampling strategy | No sampling procedure or sample size calculation was performed, as the study used the full population of disease-related publications in PubMed and complete GBD data. This comprehensive approach ensures sufficient coverage for robust, large-scale analysis. |
| Data collection | The data collection procedure relied exclusively on publicly available secondary datasets and did not involve direct interaction with human participants. We extracted disease-related publications from PubMed, using Medical Subject Headings (MeSH) from the C-branch to identify articles relevant to specific disease causes. These MeSH terms were then linked to level 2 causes in the Global Burden of Disease (GBD) database using a custom-developed large language model (LLM)–based classification tool. This tool systematically evaluated the semantic fit between each MeSH term and GBD cause to establish article–disease links. No additional individuals were present during data collection, and researcher blinding was not applicable, as the study design was purely computational and did not involve experimental conditions or human subjects. |
| Timing | Data was collected in July 2024. |
| Data exclusions | No data were excluded from the analyses. All available disease-related publications with geo-located first authors in PubMed and corresponding GBD data were included. |
| Non-participation | No participants were involved in this study; it is based entirely on secondary analysis of publicly available datasets. |
| Randomization | Please state how many participants dropped out/declined participation and the reason(s) given OR provide response rate OR state that no participants dropped out/declined participation OR if no participants were involved in the study, please state here accordingly |

# Reporting for specific materials, systems and methods

We require information from authors about some types of materials, experimental systems and methods used in many studies. Here, indicate whether each material, system or method listed is relevant to your study. If you are not sure if a list item applies to your research, read the appropriate section before selecting a response.

## Materials & experimental systems

| n/a | Involved in the study |
|-----|----------------------|
| ☒ ☐ | Antibodies |
| ☒ ☐ | Eukaryotic cell lines |
| ☒ ☐ | Palaeontology and archaeology |
| ☒ ☐ | Animals and other organisms |
| ☒ ☐ | Clinical data |
| ☒ ☐ | Dual use research of concern |
| ☒ ☐ | Plants |

## Methods

| n/a | Involved in the study |
|-----|----------------------|
| ☒ ☐ | ChIP-seq |
| ☒ ☐ | Flow cytometry |
| ☒ ☐ | MRI-based neuroimaging |

## Plants

| | |
|---|---|
| Seed stocks | *Report on the source of all seed stocks or other plant material used. If applicable, state the seed stock centre and catalogue number. If plant specimens were collected from the field, describe the collection location, date and sampling procedures.* |
| Novel plant genotypes | *Describe the methods by which all novel plant genotypes were produced. This includes those generated by transgenic approaches, gene editing, chemical/radiation-based mutagenesis and hybridization. For transgenic lines, describe the transformation method, the number of independent lines analyzed and the generation upon which experiments were performed. For gene-edited lines, describe the editor used, the endogenous sequence targeted for editing, the targeting guide RNA sequence (if applicable) and how the editor was applied.* |
| Authentication | *Describe any authentication procedures for each seed stock used or novel genotype generated. Describe any experiments used to assess the effect of a mutation and, where applicable, how potential secondary effects (e.g. second site T-DNA insertions, mosiacism, off-target gene editing) were examined.* |

