## [Peer Review File · Nature Medicine]

Global distribution of research efforts, disease burden, and impact of U.S. public funding withdrawal

Corresponding Author: Professor Marc Lerchenmueller

A version of this paper was originally rejected for publication by Nature Medicine, however that decision was reconsidered after appeal by the authors.

Version 0:

Reviewer comments:

Reviewer #1

(Remarks to the Author)

This manuscript addresses the long-standing concern regarding the alignment between scientific research efforts and the global burden of disease. The authors used a novel methodology that triangulates a large language model (LLM)-based approach, ICD-based classification, and physician input, and conducted a longitudinal analysis linking disease-specific research outputs (8.6 million articles from 1999–2021) with global disease burden (DALYs). This is very interesting. They report that while the global research-disease imbalance has seemingly improved over time, this change is almost entirely due to shifts in disease burden—specifically the decline of geographically localized communicable diseases—rather than shifts in research focus. Through the simulations they ran, the authors project a likely future reversal in the trend, with imbalance increasing again by 2050. The topic is significant given its implications for evidence-based health policy and equitable allocation of research funding as well. This study is filling a gap left by earlier cross-sectional studies, and represents a good advance through the application of LLMs to link disparate ontologies.

I have a few questions and points to raise:

- The manuscript at times implies causality or responsiveness (e.g., the “static” nature of the research enterprise), yet all findings are descriptive. No causal modeling (e.g., time-series analysis with lags or Granger causality) was undertaken to assess whether changes in disease burden precede or follow changes in research output. The notion of “responsiveness” presumes an expectation that the research enterprise should mirror disease burden dynamically. This normative assumption is not problematized or defended adequately.
- Disease areas like neoplasms, cardiovascular diseases, or neurological disorders are highly heterogeneous—treating them as uniform categories might mask nuanced shifts in subtopics (e.g., stroke vs. myocardial infarction). even though Level 2 GBD causes strike a balance between specificity and interpretability, the method involves rolling up level 3 and 4 MeSH terms.
- The reliance on affiliation strings parsed by LLM (ChatGPT) is innovative but lacks validation metrics beyond overall coverage—no inter-rater reliability or error analysis is shown for this critical step. The decision to geo-locate articles based solely on first author affiliation (even when 90% concordance with last author country is claimed) could miss important international collaborations and institutional contexts.
- DALY data is annual, but the timing of publication reflects research conducted years earlier. No lag analysis was performed to test whether research responded with delay to changes in burden.
- KLD is appropriately used and well-justified. However, KLD is sensitive to low probabilities. If any disease has a near-zero publication share but non-zero DALY share (or vice versa), the divergence can be heavily skewed. Authors could have explored smoothing or regularization.
- DALY uncertainty is incorporated via bootstrapped sampling from a normal distribution centered on the mean DALY with CI-based SDs. However, DALY distributions are typically right-skewed—not normally distributed. Assuming normality may misrepresent the true tails, underestimating variance.
- Phrases such as “static research enterprise” or “unresponsive research” carry evaluative weight. These need qualification, as the expectation that research should precisely mirror DALYs is debated.

All in all, I find the study strong and innovative. The authors have done a wonderful job achieving 94.9% accuracy at the article level in linking MeSH terms with disease burden categories. They used a well-representative dataset of over 8.5

million papers. Extensive global geographic coverage. Good use of KLD. Integration of funding information for about 40% of the papers. Successful geolocation of research outputs, mapping 89% of articles. Once the above mentioned points are addressed, I think this paper brings good value to readers of Nature Medicine.

Reviewer #2

(Remarks to the Author)

Dear authors,

This is very interesting manuscript with a robust methodology and 8.6 million disease-specific research articles.

You write: "Central to our assessment of imbalance is the normative principle that the distribution of research should follow the distribution of disease burden."

As you are well aware, much health/medical research is driven by the pharmaceutical industry, especially as they have drugs in Phase 3 trials or approved. The paper did not address this important point fully, though you do state: "we obtain no evidence that funding would have materially influenced the direction of research in line with changes in the burden of disease". Please clarify. And given the ongoing changes at the NIH, the world's largest health research funder, you may want to discuss what impact this will have on your findings.

Also, given the lag time between disease burden and funding, related to anticipation of disease prevalence (for ex the growth in NCDs in Africa) and legacy funding (HIV) coupled with "latency", could you consider an expected interval that would be "reasonable" for research and BoD to better align when presenting the results (eg number of years)?

I also think it could be interesting for readers to see the classic GBD table of BoD by disease, the change (with arrows) over the time period you studied, and something similar with regards to the diseases/conditions you studied. It makes the disconnect you found even more easier to visualise. You might also group research into "clinical," "public health," "health systems" etc. I note that you did not mention the latter at all, research areas that are traditionally underfunded.

Finally, since you mention a need for global monitoring and possibly coordination (see suggestions below), you may want to mention and cite the new WHO Global Health Strategy, which in theory guides the world in the coming years: https://cdn.who.int/media/docs/default-source/documents/about-us/general-programme-of-work/global-health-strategy-2025-2028.pdf?sfvrsn=237faeeb_3#:~:text=At its heart is a,access to health services and

In the Abstract you write "We conclude with policy recommendations to..." but it is difficult to find the policy recommendations. Can you consider a box or listing them so a reader can easily find the policy recommendations aligned with your conclusion: "this study underscores the critical need for more adaptive and equitable science policies to address persistent disparities in global research"?

Ref 9 can be updated, noting that newer GBD studies will include the COVID-19 pandemic era.

Re the NIH, I think the increase in budget was not just related to health, but also wealth (often for the US), with patents developed, drug discovery, jobs created etc. Since the reference is from 2005, perhaps a more nuanced view could be included and referenced.

p. 2 Should middle-income countries be included along with LIC? The reference (#2) is a bit old and perhaps large countries like China and India, now MIC were LIC then?

Alternatively, note that hundreds of millions of very poor people living in MIC are also affected.

Discussion: Please clarify "if the research enterprise remains as rigid in its allocation of research as it has been over the past two decades,". It reads as if research funding is allocated centrally, when although there are some very large funders (NIH, European Commission, Wellcome Trust...), I don't think there is central or even coordinated funding decisions. And the, as mentioned, there is the large private sector investments in research, from drug development to drug roll-out, HEOR, etc.

Discussion: The call for a "global monitoring body" was made in 2013. Why do you think it has not happened since and any suggestions on the structure? You might want to comment on the Global Forum for Health Research (GFHR) and the Council on Health Research for Development (COHRED)

Discussion. Where you write "we invite researchers to explore" you may want to call out health decisionmakers at the national and international levels too.

In the Disc, before concluding that there is a "critical need for more adaptive and equitable science policies to address persistent disparities in global research" can the authors provide some concrete recommendations?

In Fig 1B, change the colour of Greenland to that of Denmark. Greenland is an autonomous territory in the Kingdom of Denmark. This comment has nothing to do with the current US policy towards Greenland.

Maybe the first line of the Methods is better for the Disc "To link time-varying global burden of disease data with corresponding disease-specific research publications in the life sciences, we developed a novel approach to construct a

foundational dataset that integrates multiple databases."

Reviewer #3

(Remarks to the Author)

Dear Editor,

Thanks for inviting me to review the interesting manuscript titled 'A static research enterprise decouples from changes in the burden of disease'. The authors addressed an important issue: the misalignment of research priorities, volume, and evolving disease burden over time. The manuscript is well-written, and the author has used appropriate statistical methods. However, some minor concerns, listed below, need to be addressed before publication.

1. The current title is less readable- some readers may find it challenging to understand what this means. Given the comprehensive nature of the study, encompassing research volume in the broad health sector, I would suggest revising the title.
2. (Abstract) Like the title, the abstract is also not readable to a lay audience. The inclusion of the references makes it more like an introduction. Many readers may not turn out to read the whole manuscript if they find the abstract challenging. I would suggest revising the background sentences without references and clearly stating the study's objectives.
3. Methods (Page: 25): The authors assigned the publications to geographical regions based on the affiliation information associated with the first authors in the bylines of the research articles. A substantial proportion of research conducted in low-income countries, especially in the sub-Saharan African region and Asia, often has the first author's affiliation with developed countries. Given that low-income countries are a big contributor to the global disease burden and have limited resources for improving the research enterprise rather than fighting against the disease burden, is the LLM model capable of identifying the geographic region based on the research setting, in addition to the research enterprise?
4. Methods (Page 26): The authors used 95% CI for KLD. While the GBD DALY estimates were accompanied by an uncertainty level (UL), would it be beneficial to estimate the UL rather than the confidence interval (CI) to incorporate multiple sources of uncertainty?
5. Figure 2 (page 16): The Figure A caption is somewhat confusing, as it's unclear whether it applies to all 16 level 2 diseases combined or a particular one. It would be great to see how each disease level looks over time separately, perhaps combined here and for each disease level in the supplementary. Also, the DALY for some disease levels (communicable disease as well as maternal and neonatal) had a more significant decline over the last two decades, whereas an increase for some communicable diseases, so separate for disease level as well as geographic region level may be useful.
6. Results (page 9-10) or Figure 6 (20): As of comments five, the dynamics of the imbalance between research output and disease burden differ by disease level and region, so while projecting for next decades or so require a range of assumptions, which are not mentioned in the methods section. It's important to highlight the variations when providing a global projection for all disease levels combined.
7. Additionally, the DALY estimation process in the Global Burden of Disease has evolved over the last two decades and is expected to continue changing in the coming years with the incorporation of new risk factors and diseases. However, the research enterprise outcome account is a static process. While the authors acknowledge the limitation of the DALY and mention that it's the best available health burden measure, did the authors consider some weighting method to mitigate the bias? For example, providing weight in accordance with the evolving nature of the DALY estimates.
8. I'm not sure about the journal structure; currently, the introduction and results sections are difficult to separate. Providing the section title 'Results' before presenting the results may be helpful for readers.

Version 1:

Reviewer comments:

Reviewer #1

(Remarks to the Author)

Thank you for the chance to see the revised version of this work. I do see many improvements in this updated manuscript, and I thank the authors for their revisions and answers to my comments/questions.

The authors seem to have addressed several of the concerns I raised well, including:

- language and framing that gives the impression of causality; glad to see this ameliorated
- Glad to see the new lagged analysis, and I find its results interesting
- Also happy to see more compelling Discussion of the results with some of the changes the authors made.
- Great that you also now address the changes in research funding in the US
- Thank you for adding the inter-rater metrics and validation for the LLM parsing of the affiliations, and kudos for now including affiliations of all authors
- Thank you for avoiding the normality assumption with DALY uncertainties and bootstrapped sampling

I have no further comments or questions for the authors.

Reviewer #2

(Remarks to the Author)

Thank you for the strong, detailed responses to each point I and the other reviewers made.

It was very interesting to read how you addressed the points and the changes made in the manuscript. I have no further comments.

Reviewer #3

(Remarks to the Author)

I appreciate the authors for updating the manuscript accordingly. I'm happy with the current version to be accepted for publication in Nature Medicine.

Point-by-point response

Your Article entitled "A static research enterprise decouples from changes in the burden of disease" has now been seen by 3 referees, whose comments are attached. While they find your work of potential interest, they have raised serious concerns which in our view are sufficiently important that they preclude publication of the work in Nature Medicine, at least in its present form.

Should further analyses and clarifications allow you to fully address the editorial and peer reviewers' points we would be willing to consider an appeal of our decision (unless, of course, something similar has by then been accepted at Nature Medicine or appeared elsewhere). This includes submission or publication of a portion of this work someplace else.

We appreciate your willingness to consider an appeal of the decision and possible consideration of a revised manuscript. We are enclosing a separate letter of appeal, a revised manuscript and supplement, and we are addressing each of the comments raised by the editor and reviewers in this point-by-point response. The received comments are in black, our responses in blue.

The required clarifications include; discussion of how the study is relevant in relation to recent changes in changes to the landscape of cuts to research funding and international aid, why the study should focus solely on burden of disease by DALYs and how this would be useful, and highlighting the direct, actionable broader policy recommendations to policymakers, funders and other stakeholders stemming from these findings. Including a policy summary table highlighting the key policy aspects stemming from the work would be important to communicate these messages (see an example here: <https://www.nature.com/articles/s41591-023-02600-4/tables/1>). We hope you understand that until we have read the revised manuscript in its entirety we cannot promise that it will be sent back for peer review.

We have extensively revised our manuscript and additionally added new data and analyses to address all your highlighted clarifications. In summary:

1. We have added a simulation of the prospective divergence between the global research enterprise and the burden of disease, projecting the consequences of a withdrawal of United States (U.S.) public funding for researchers affiliated outside the U.S. and find:
 - 1.1. The divergence between research and global disease burden would increase abruptly as a result of a withdrawal of U.S. public funding.
⇒ **Please see new Figure 6A, Results Section page 10-11 and Discussion Section page 13-14**
 - 1.2. To locate the sources of the projected increase in divergence, we have additionally created a heat map across diseases and geographic regions to concretely visualize the level of exposure to U.S. public research funding. Our results show that specific communicable diseases (e.g., HIV/Aids, respiratory infections and tuberculosis) as well as certain non-communicable diseases (e.g., neurological disorders and substance use disorders) are at elevated risk from a U.S. retrenchment. In terms of geography, research conducted in Sub-Saharan Africa is distinctly exposed to U.S. financial support. We also discuss the policy implications from these findings in detail.
⇒ **Please see new Figure 6B, Results Section page 11 and Discussion Section page 13-15**

- 1.3. Finally, we examined which other countries might compensate in the short term for a funding gap created by the potential withdrawal of U.S. public funding. Assuming that existing high levels of public funding in a given country approximate the resource base that can be scaled in the absence of U.S. public funding, China, the UK, Germany, Canada, and Australia emerge as the most likely countries to step in, albeit for different diseases (e.g., China for respiratory infections and TB, the UK for HIV/AIDS, Canada and Australia for substance use disorders).
 - ⇒ **Please see new Table 1, Supplementary Figure 7, Results Section page 11 and Discussion Section page 13**
2. To broaden our assessment of disease burden beyond DALYs, we now include an additional analysis of the research-disease divergence using also mortality ("deaths") and prevalence as additional health outcome measures. Our findings of a declining divergence between research and disease burden remain consistent across the measures of prevalence, deaths, and DALYs. We have also added more explanation about why we focus our analyses on DALYs as a measure of the global burden of disease, including the integrative assessment of morbidity and mortality inherent in DALYs and the explicit conception of DALYs as an instrument to inform evidence-based agenda setting.
 - ⇒ **Please see new Supplementary Figures 2A, 2B, and 2C, Results Section page 6, Discussion Section page 15**
3. We have now included a new Table 1 that summarizes our key findings and links them to direct, actionable, broader policy recommendations to policymakers, funders and other stakeholders, including industry. We elaborate on these recommendations in an expanded discussion section. We have also stated more clearly in the introduction and discussion sections that evidence-based recommendations cannot be reliably drawn without a longitudinal assessment of progress. We highlight how the novel data platform offered through this work for the first time can help stakeholders in ongoing monitoring and agenda setting.
 - ⇒ **Please see new Table 1, Introduction Section page 2-3, revised Discussion Section**

Reviewers' Comments:

Reviewer #1 (Remarks to the Author):

This manuscript addresses the long-standing concern regarding the alignment between scientific research efforts and the global burden of disease. The authors used a novel methodology that triangulates a large language model (LLM)-based approach, ICD-based classification, and physician input, and conducted a longitudinal analysis linking disease-specific research outputs (8.6 million articles from 1999–2021) with global disease burden (DALYs). This is very interesting. They report that while the global research-disease imbalance has seemingly improved over time, this change is almost entirely due to shifts in disease burden—specifically the decline of geographically localized communicable diseases—rather than shifts in research focus. Through the simulations they ran, the authors project a likely future reversal in the trend, with imbalance increasing again by 2050. The topic is significant given its implications for evidence-based health policy and equitable allocation of research funding as well. This study is filling a gap left by earlier cross-sectional studies, and represents a good advance through the application of LLMs to link disparate ontologies.

Thank you for highlighting the significance of our work and the gaps filled. Your constructive feedback tremendously helped us to further improve our work. Please see below our responses to each of your points raised.

I have a few questions and points to raise:

- The manuscript at times implies causality or responsiveness (e.g., the “static” nature of the research enterprise), yet all findings are descriptive. No causal modeling (e.g., time-series analysis with lags or Granger causality) was undertaken to assess whether changes in disease burden precede or follow changes in research output. The notion of “responsiveness” presumes an expectation that the research enterprise should mirror disease burden dynamically. This normative assumption is not problematized or defended adequately.

R1.1. We have undertaken several steps to address this very important point you raise.

First, we have revised our language throughout the manuscript to mitigate the impression of causal claims. In particular, we now consistently use the term “divergence”, as opposed to e.g., imbalance or deviance, to describe a more neutral assessment of the comparison between the distribution of research and disease burden.

⇒ **Please see new Title and consistent use of the concept of research-disease divergence**

Second, in response to your comment here and your related comment below (R1.4), we now provide an additional time-lagged sensitivity analysis of the declining research-disease divergence (KLD). Specifically, we compare the distribution of DALYs in a given year with the distribution of research 10 years later. To make this lagged analysis possible, we extended our DALY data coverage by an additional decade (1990–1999). The earliest comparison calculates the KLD between the 1990 DALY distribution and the 2000 research distribution, while the latest comparison calculates the KLD between the 2009 DALY distribution and the 2019 research distribution, spanning 30 years of data (1990-2019). This lagged analysis yields no evidence that research adapts to changes in the burden of disease, either contemporaneously or with a lag of up to 10 years. To our minds, these results suggest that, while the direction of causality is certainly of theoretical interest, the empirical stability of the research distribution over the course

of many years during which disease burden changed allow for an unambiguous interpretation of the key finding from our work: The decline in the research-disease divergence is almost entirely attributable to changes in the distribution of disease burden. Throughout the main manuscript, we are careful to avoid implying causality and consistently frame our findings as descriptive differences in the temporal distributions of disease burden and research focus.

⇒ **Please see new Supplementary Figure 4, Results Section page 7**

Third, we now open the discussion section with a problematization of divergence between research and disease burden, weighing arguments for greater convergence against (a) conceivably indicated areas for divergence and (b) against the apparent status quo of a largely stable research enterprise in a dynamically changing disease landscape.

⇒ **Please see Discussion section pages 12–13**

- Disease areas like neoplasms, cardiovascular diseases, or neurological disorders are highly heterogeneous—treating them as uniform categories might mask nuanced shifts in subtopics (e.g., stroke vs. myocardial infarction). even though Level 2 GBD causes strike a balance between specificity and interpretability, the method involves rolling up level 3 and 4 MeSH terms.

R1.2. For the scope and objectives of the current study, we chose Level 2 of the GBD hierarchy, exactly for the reason you mention: it provides the best achievable granularity while maintaining a high degree of specificity. Still, we have now addressed the potential heterogeneity within cause categories, which might involve nuanced shifts not captured in this analysis, by explicitly noting this limitation in the manuscript. In addition, we offer a sensitivity analysis using Level 3 disease categories of the GBD hierarchy. This analysis yields results consistent with our main findings at Level 2 – specifically, a reduction in imbalance between disease burden and research over time driven by changes in DALYs rather than changes in research output. Furthermore, we selected the two major Level 2 research areas (cardiovascular disease and neoplasms) as examples to show how the underlying research and disease burden at Level 3 are distributed. The patterns at Level 3 closely mirror those observed at Level 2: research on neoplasms remains disproportionately high relative to their burden, while research on major cardiovascular diseases continues to lag behind their burden. This consistency further strengthens the robustness of our findings across different levels of disease categorization. We highlight that future studies will delve deeper into specific causes and geographic variations, among other more granular research questions.

⇒ **Please see new Supplementary Figures 8A, 8B, 8C, and Discussion Section page 15-16**

- The reliance on affiliation strings parsed by LLM (ChatGPT) is innovative but lacks validation metrics beyond overall coverage—no inter-rater reliability or error analysis is shown for this critical step.

R1.3. To validate the affiliation strings parsed with the LLM, we include an analysis with our revision for which we randomly selected 200 affiliation strings and had two independent raters evaluate the country designations produced by the LLM. In all cases (100%), both raters confirmed that the LLM's country assignments were correct.

⇒ **Please see Methods Section page 30**

The decision to geo-locate articles based solely on first author affiliation (even when 90% concordance with last author country is claimed) could miss important international collaborations and institutional contexts.

R1.4. As part of the revision, we have now designated the affiliations of all authors – first, last, and interior – and included sensitivity analyses across regions. We find that the relative proportions of authorship vary only slightly when comparing designations based on any authorship position versus first authorship alone. However, our new data suggest that non-first authorships may play a greater role in certain regions, such as East and Southeast Asia. Future research could explore regional variations in research-disease divergence in more detail, with particular attention to these patterns.

⇒ **Please see new Supplementary Figure 6B, Methods Section page 30**

- DALY data is annual, but the timing of publication reflects research conducted years earlier. No lag analysis was performed to test whether research responded with delay to changes in burden.

R1.5. In response to this comment and our earlier response to R1.1, we now extend our analysis of DALY distributions back to 1990 (compared to the original starting point of 1999) and compare them to research output lagged by 10 years. Specifically, we calculate the Kullback-Leibler divergence (KLD) annually, comparing the 1990 DALY distribution to 2000 research output, the 1991 DALY distribution to 2001 research output, and so on, through to the comparison of 2009 DALYs with 2019 research. Our results show no significant adaptation of research to changes in disease burden, even with a 10-year lag, indicating that shifts in DALY distributions rather than changes in research output drive the observed reduction in divergence.

⇒ **Please see new Supplementary Figure 4, Results Section page 7**

- KLD is appropriately used and well-justified. However, KLD is sensitive to low probabilities. If any disease has a near-zero publication share but non-zero DALY share (or vice versa), the divergence can be heavily skewed. Authors could have explored smoothing or regularization.

R1.6. We now calculate new sensitivity analyses to address this point. First, we recalculate the longitudinal evolution of the KLD, excluding near-zero outliers (0.01) from the numerator and denominator. Second, we perform Laplace smoothing, adding a constant alpha of 0.1% and 0.5%. We obtain consistent results.

⇒ **Please see Methods Section page 31**

- DALY uncertainty is incorporated via bootstrapped sampling from a normal distribution centered on the mean DALY with CI-based SDs. However, DALY distributions are typically right-skewed—not normally distributed. Assuming normality may misrepresent the true tails, underestimating variance.

R1.7. To address this point, we now draw the bootstrap samples from a log-normal distribution, which better captures the positive skew and non-negative nature of DALYs compared to a normal distribution. We apply this log-normal measure of uncertainty consistently throughout the revised manuscript.

⇒ **Please see Methods Section page 31**

- Phrases such as “static research enterprise” or “unresponsive research” carry evaluative weight. These need qualification, as the expectation that research should precisely mirror DALYs is debated.

R1.8. We have revised the language throughout the manuscript to avoid phrases that might imply evaluative judgments. In addition, we have taken special care to ensure that our policy recommendations remain descriptive rather than prescriptive. For example, we have consistently used the term “divergence” to describe the comparison of research with disease burden distributions, which is also more consistent with our KLD measure. Also, we open our discussion section with an encompassing weighting of convergence, divergence, and stability.

⇒ **Please see e.g., Discussion Section page 12-13**

All in all, I find the study strong and innovative. The authors have done a wonderful job achieving 94.9% accuracy at the article level in linking MeSH terms with disease burden categories. They used a well-representative dataset of over 8.5 million papers. Extensive global geographic coverage. Good use of KLD. Integration of funding information for about 40% of the papers. Successful geolocation of research outputs, mapping 89% of articles. Once the above mentioned points are addressed, I think this paper brings good value to readers of Nature Medicine.

R1.9. Thank you again for your constructive feedback. Incorporating your suggestions has significantly strengthened our work, and we hope we have fully addressed your comments.

Reviewer #2 (Remarks to the Author):

Dear authors,

This is very interesting manuscript with a robust methodology and 8.6 million disease-specific research articles.

Thank you for highlighting the interesting nature of our work. Your constructive feedback tremendously helped us improve our manuscript. Please see below our responses to each of your points raised.

- You write: "Central to our assessment of imbalance is the normative principle that the distribution of research should follow the distribution of disease burden."
As you are well aware, much health/medical research is driven by the pharmaceutical industry, especially as they have drugs in Phase 3 trials or approved. The paper did not address this important point fully, though you do state: "we obtain no evidence that funding would have materially influenced the direction of research in line with changes in the burden of disease". Please clarify.

R2.1. We agree with you that the research and development investments by industry are not fully captured in our funding data that is created from funding acknowledgements. We state this clearly in our limitations in the discussion section. We also point out the importance of including industry as a critical stakeholder in future efforts targeting better alignment of research and disease.

Beyond these important modifications of the manuscript, we also leverage our data to analyze two additional subsamples that speak to the role of industry. First, we identify all authors with an industry affiliation and isolate the papers with at least one industry-affiliated author. This may not only serve as an approximation of the distribution of research interests within industry but may also capture certain levels of funding, even if not explicitly acknowledged. We obtain no differences in this subsample relative to our complete sample of research.

Next, we specifically focus on the identification of Phase 3 clinical trials. To that end, we have incorporated new data that links publications in our dataset to registered trials in ClinicalTrials.gov. This linkage allows us to identify publications reporting on clinical trials and extract additional metadata – most importantly, the sponsor and the phase of the clinical trial. Please note that NCT IDs have been consistently available in the PubMed XML dataset only since 2009. For Phase 3 trials with industry sponsors we obtain a significant increase in research-disease divergence and we interpret this finding to support our recommendation for greater industry involvement in the coordination of the research enterprise.

⇒ **Please see new Supplementary Figure 5C, Results Section pages 7, Discussion Section e.g., page 15**

- And given the ongoing changes at the NIH, the world's largest health research funder, you may want to discuss what impact this will have on your findings.

R2.2. We have extensively revised our manuscript to add new data, discussion, and a table summarizing our findings and policy recommendations regarding ongoing changes at the NIH and other major public U.S. funders of health research. Specifically, we include a simulation in our forecast until 2050 that models the prospective divergence between the global research enterprise and the burden of disease under a hypothetical withdrawal of U.S. public funding for

researchers affiliated outside the U.S. Our analysis shows that such a withdrawal would sharply exacerbate the divergence between research and global disease burden.

This motivated us to examine which areas are most vulnerable to changes in US public funding. We find that work conducted in Sub-Saharan Africa would be particularly affected. In terms of diseases, both communicable diseases (e.g., HIV/Aids, respiratory infections and tuberculosis) as well as certain non-communicable diseases (e.g., neurological disorders and substance use disorders) are at elevated risk from a U.S. retrenchment.

To further assess the global impact, we explored which other countries might have the capacity to fill a potential funding gap. Focusing on the five diseases most reliant on US public funding, we identified countries with already high levels of investment as the most likely candidates to scale up support. These findings underscore the global consequences of shifts in U.S. public funding priorities and reinforce the need for international coordination and contingency planning in health research investment.

⇒ Please see new Figure 6A/B, Table 1, Supplementary Figure 7, Results Section pages 9-11, Discussion Section pages 14-15

- Also, given the lag time between disease burden and funding, related to anticipation of disease prevalence (for ex the growth in NCDs in Africa) and legacy funding (HIV) coupled with "latency", could you consider an expected interval that would be "reasonable" for research and BoD to better align when presenting the results (eg number of years)?

R2.3. We agree that the alignment between research and disease burden is influenced by both anticipatory dynamics (e.g., projected NCD growth) and legacy effects (e.g., sustained HIV funding), as well as inherent research latency. While we refrain from defining a single "reasonable" interval given variation across fields and funding systems, prior literature suggests that research impact typically unfolds over a 10–20 year horizon (e.g., Ref #40,41).

To assess this legacy effect, we conducted new analyses comparing DALY distributions with research output lagged by 10 years. Using extended DALY data from 1990 onward, we calculated the Kullback-Leibler Divergence between, for example, 1990 DALYs and 2000 research, and between 2009 DALYs and 2019 research. These analyses show no evidence that research aligns with changes in disease burden, even with a decade-long lag. Given the consistency of research patterns, similar results would be expected for longer time lags. Overall, this analysis supports our main finding of the change in research-disease divergence driven by changes in disease burden.

⇒ Please see new Supplementary Figure 4, Results Section page 7

- I also think it could be interesting for readers to see the classic GBD table of BoD by disease, the change (with arrows) over the time period you studied, and something similar with regards to the diseases/conditions you studied. It make the disconnect you found even more easier to visualise. You might also group research into "clinical," "public health," "health systems" etc. I note that you did not mention the latter at all, research areas that are traditionally underfunded.

R2.4. We have now added a Table to further contextualize our results using the classic GBD table format, showing that while the relative ranking of disease burden changed quite a bit over the past 20-years, there is minimal change in the relative ranking of disease-specific research.

As per your recommendation, we also isolated research publications from public health journals and health systems journals based on the Clarivate Journal Citation Report journal categories. As you suspect, these disciplines are underrepresented (less than 4% of publications fall into these categories in our dataset), while only public health research seems more closely aligned with the distribution with the global burden of disease.

⇒ **Please see new Supplementary Figures 3G and 5D**

- Finally, since you mention a need for global monitoring and possibly coordination (see suggestions below), you may want to mention and cite the new WHO Global Health Strategy, which in theory guides the world in the coming years: https://cdn.who.int/media/docs/default-source/documents/about-us/general-programme-of-work/global-health-strategy-2025-2028.pdf?sfvrsn=237faeeb_3#:~:text=At its heart is access to health services.

R2.5. In response to your comment, we have now cited the WHO Global Health Strategy (2025–2028) and highlighted its central goal of ensuring access to health services as a foundation for global health efforts. We have also elaborated, within the constraints of the word limit, on how our database and methodological approach can contribute to the strategy's implementation, particularly by enabling improved global monitoring of research–disease alignment and supporting more coordinated responses to health priorities.

⇒ **Please see Discussion Section pages 14-16**

- In the Abstract you write "We conclude with policy recommendations to..." but it is difficult to find the policy recommendations. Can you consider a box or listing them so a reader can easily find the policy recommendations aligned with your conclusion: "this study underscores the critical need for more adaptive and equitable science policies to address persistent disparities in global research"?

R2.6. We have extensively revised our discussion section to make the policy recommendations more visible and accessible. Specifically, we have added a clearly labeled table that lists our main policy recommendations in a concise and structured format. We believe this addition will help readers quickly identify the key takeaways and better understand their relevance to different stakeholder groups.

⇒ **Please see new Table 1, revised Discussion Section**

- Ref 9 can be updated, noting that newer GBD studies will include the COVID-19 pandemic era.

R2.7 We have updated the reference to the most recent GBD study that encompasses data from 1990 – 2021, as used in our study.

⇒ **Please see updated reference #2**

- Re the NIH, I think the increase in budget was not just related to health, but also wealth (often for the US), with patents developed, drug discovery, jobs created etc. Since the reference is from 2005, perhaps a more nuanced view could be included and referenced.

R2.8 We now include a more nuanced reflection of the NIH budget increase, noting that it has been driven not only by public health priorities but also by broader economic considerations that also materialized. We have also incorporated more recent and relevant references that highlight the multifaceted effects, beyond the 2005 citation used in the earlier version.

⇒ **Please see updated Introduction section page 2 and new reference #15-19**

- p. 2 Should middle-income countries be included along with LIC? The reference (#2) is a bit old and perhaps large countries like China and India, now MIC were LIC then?
Alternatively, note that hundreds of millions of very poor people living in MIC are also affected.

R2.9 Yes, we corrected our sentence to include middle-income countries and added a more recent publication that investigates the connection between scientific and economic development.

⇒ **Please see updated Introduction section page 2 and references #5,24**

- Discussion: Please clarify "if the research enterprise remains as rigid in its allocation of research as it has been over the past two decades,". It reads as if research funding is allocated centrally, when although there are some very large funders (NIH, European Commission, Wellcome Trust...), I don't think there is central or even coordinated funding decisions. And the, as mentioned, there is the large private sector investments in research, from drug development to drug roll-out, HEOR, etc.

R2.10. We have rephrased this sentence in the revised manuscript to avoid implying a centrally coordinated research funding system. To the contrary, we now clarify that while major public funders play a significant role, research funding remains fragmented and uncoordinated, particularly across sectors and regions. In response to your comment, as well as your points R2.1, R2.5, and R2.11, we have elaborated on the need for greater collaboration and more systematic monitoring of research investments, including those from industry. These points have been integrated into the main discussion.

⇒ **Please see Results Section page 7, Discussion Section pages 12-15**

- Discussion: The call for a "global monitoring body" was made in 2013. Why do you think it has not happened since and any suggestions on the structure? You might want to comment on the Global Forum for Health Research (GFHR) and the Council on Health Research for Development (COHRED)

R2.11 We have elaborated on the termination of previous coordination attempts and specifically reference the GFHR and COHRED.

⇒ **Please see Discussion Section page 14**

- Discussion. Where you write "we invite researchers to explore" you may want to call out health decisionmakers at the national and international levels too.

R2.11 In the revised manuscript, we have expanded the discussion to explicitly address health decisionmakers at both national and international levels, in addition to researchers. We now emphasize the importance of their engagement in addressing the disparities highlighted by our study.

⇒ **Please see Discussion Section page 15**

- In the Disc, before concluding that there is a "critical need for more adaptive and equitable science policies to address persistent disparities in global research" can the authors provide some concrete recommendations?

R2.12 We have substantially revised the manuscript, with particular focus on the discussion section, to directly address the call for more clearly articulated recommendations. In response to

your suggestion, we now include specific, actionable proposals aimed at fostering more adaptive and equitable science policies – particularly in relation to funding allocation, capacity building, and international collaboration. These recommendations are framed in light of persistent global research disparities and recent political developments in the United States. As noted in our response to point R2.2, we have also added a summary table that presents the key findings of our study alongside the corresponding policy implications.

⇒ **Please see new Figure 6A/B, Table 1, revised Discussion Section**

- In Fig 1B, change the colour of Greenland to that of Denmark. Greenland is an autonomous territory in the Kingdom of Denmark. This comment has nothing to do with the current US policy towards Greenland.

R2.13. Thank you for the thorough review and catching this mislabeling. We have corrected the coloring of Figure 1B to reflect Greenland's status as an autonomous territory of Denmark.

⇒ **Please see revised Figure 1B**

- Maybe the first line of the Methods is better for the Disc "To link time-varying global burden of disease data with corresponding disease-specific research publications in the life sciences, we developed a novel approach to construct a foundational dataset that integrates multiple databases."

R2.14. We followed your recommendation and removed the sentence from the methods section and state the novelty and methodological contribution in the discussion section.

⇒ **Please see updated Methods and Discussion Section.**

Reviewer #3 (Remarks to the Author):

Dear Editor,

Thanks for inviting me to review the interesting manuscript titled 'A static research enterprise decouples from changes in the burden of disease'. The authors addressed an important issue: the misalignment of research priorities, volume, and evolving disease burden over time. The manuscript is well-written, and the author has used appropriate statistical methods. However, some minor concerns, listed below, need to be addressed before publication.

Thank you for emphasizing the importance of our work. Your constructive feedback has been invaluable in helping us improve our manuscript. Please find below our responses to each of the points you raised.

1. The current title is less readable- some readers may find it challenging to understand what this means. Given the comprehensive nature of the study, encompassing research volume in the broad health sector, I would suggest revising the title.

R3.1. Thank you for your positive assessment of our study and for pointing out the importance of a crisp title that makes the content more accessible to a broad readership. We have now revised the title.

⇒ **Please see revised Title**

2. (Abstract) Like the title, the abstract is also not readable to a lay audience. The inclusion of the references makes it more like an introduction. Many readers may not turn out to read the whole manuscript if they find the abstract challenging. I would suggest revising the background sentences without references and clearly stating the study's objectives.

R.3.2. We have revised the abstract in accordance with your recommendations. Additionally, we have added a new Table 1 to succinctly summarize the study's objective, main findings, and policy implications in a way that is accessible to a broad readership.

⇒ **Please see revised Abstract and new Table 1**

3. Methods (Page: 25): The authors assigned the publications to geographical regions based on the affiliation information associated with the first authors in the bylines of the research articles. A substantial proportion of research conducted in low-income countries, especially in the sub-Saharan African region and Asia, often has the first author's affiliation with developed countries. Given that low-income countries are a big contributor to the global disease burden and have limited resources for improving the research enterprise rather than fighting against the disease burden, is the LLM model capable of identifying the geographic region based on the research setting, in addition to the research enterprise?

R3.3. The objective of our study was to associate articles geographically based on the localization of the authors, rather than the research setting itself. To move beyond a focus solely on first authors, we have now also incorporated information on interior and last authorships, as suggested by another reviewer, and find similar results.

To address the question of whether the LLM model could also analyze the research setting – an important and interesting point – we randomly selected 500 articles from our dataset and manually retrieved their abstracts. Abstracts were successfully obtained for 435 of these articles, and information on the study location could be extracted from 71 abstracts.

A key limitation appears to be the frequent lack of research setting information in abstracts; this issue could potentially be mitigated if full-text searches were feasible on a large scale. Nevertheless, in 60 of the 71 cases (85%), the first author's country matched one of the study locations extracted from the abstract. In other words, in only 11 out of the 500 sampled articles (2.2%), the abstract contained study location information that differs from the geography of the first author's affiliation.

⇒ **Please see new Supplementary Figure 6B**

4. Methods (Page 26): The authors used 95% CI for KLD. While the GBD DALY estimates were accompanied by an uncertainty level (UL), would it be beneficial to estimate the UL rather than the confidence interval (CI) to incorporate multiple sources of uncertainty?

R3.4. Thank you for your comment pointing out that the DALY estimates from the Global Burden of Disease (GBD) study come with non-symmetric uncertainty intervals, and that using these uncertainty levels (ULs) directly, rather than estimating symmetric confidence intervals (CIs), can better reflect the multiple sources of uncertainty inherent in the DALY data.

In response, we have revised our approach to better incorporate the uncertainty inherent in the DALY data. Specifically, rather than relying on symmetric confidence intervals based on a normal distribution, we now make use of the non-symmetric upper and lower bounds (UL and LL) reported by the Institute for Health Metrics and Evaluation (IHME). We model the DALY estimates using a log-normal distribution, which allows us to reflect the skewed and non-negative nature of the data while aligning the modeled distribution with the reported uncertainty bounds. This change enables us to more accurately capture the uncertainty structure of the DALY estimates, which implicitly includes multiple sources of uncertainty as accounted for by the IHME. We now report uncertainty intervals for KLD that are derived from this updated approach.

⇒ **Please see updated Methods Section page 31**

5. Figure 2 (page 16): The Figure A caption is somewhat confusing, as it's unclear whether it applies to all 16 level 2 diseases combined or a particular one. It would be great to see how each disease level looks over time separately, perhaps combined here and for each disease level in the supplementary. Also, the DALY for some disease levels (communicable disease as well as maternal and neonatal) had a more significant decline over the last two decades, whereas an increase for some communicable diseases, so separate for disease level as well as geographic region level may be useful.

R3.5. We have revised the caption of Figure 2A to clarify that the research-disease divergence (KLD) is calculated using aggregate data across all 16 Level 2 disease categories. In response to your suggestion for greater granularity, we have added two new longitudinal figures to the Supplementary Materials, one showing global trends in research output and DALYs over time, disaggregated by each of the 16 disease categories and one extending this analysis by presenting the same disease-specific trends separately for each world region. We did not disaggregate by disease and region simultaneously in a single figure, as this would produce 112 combinations (16 disease categories × 7 regions), making interpretation difficult. These additions allow for a more detailed exploration of the temporal patterns you highlighted, such as the marked decline in DALYs for communicable, maternal, and neonatal conditions, as well as increases in certain non-communicable diseases.

⇒ **Please see new Supplementary Figures 3A, 3B, and 3C**

6. Results (page 9-10) or Figure 6 (20): As of comments five, the dynamics of the imbalance between research output and disease burden differ by disease level and region, so while projecting for next decades or so require a range of assumptions, which are not mentioned in the methods section. It's important to highlight the variations when providing a global projection for all disease levels combined.

R3.6. We agree with you, the research-disease divergence varies across disease category and world region. In the original version of the manuscript, our forecasting model was based on global aggregates and assumed linear continuity of pre-pandemic trends in both research output and disease burden. We also agree that we had not sufficiently described these assumptions in the methods section. Your comment prompted us to revisit and refine our approach. In re-examining the literature in the course of our revision, we found a recent *Lancet* publication that provides comprehensive forecasts of DALY estimates for 204 countries and territories through 2050 (Vollset, S. E., Ababneh, H. S., Abate, Y. H., et al. (2024). *Burden of disease scenarios for 204 countries and territories, 2022–2050: a forecasting analysis for the Global Burden of Disease Study 2021*. *The Lancet*, 403(10440), 2204–2256).

We have now integrated these data into our forecasting model to generate future DALY distributions. Similar to our initial approach, these data still assume that future changes will generally follow historical progress. However, using the data from the Lancet study allows us to base our DALY projections on empirically grounded, peer-reviewed forecasts, which turned out to be quite similar to our originally submitted forecast. For projection the evolution of research, we continue to assume a linear trend, as the global research enterprise tends to evolve slowly and incrementally, as we document throughout our analyses. Its institutional stability, long funding cycles, and cumulative nature make this assumption plausible at the aggregate level. We have revised the methods section to more clearly articulate these assumptions, data sources, and their limitations. To further illustrate these assumptions and address your point about variation, we have added a new supplementary figure showing the projected DALY shares and research output shares by disease category. This addition helps contextualize the global-level projection presented in Figure 6 of the main manuscript.

⇒ **Please see updated Methods Section pages 32-33, new Supplementary Figures 3D, 3E, and 3F**

7. Additionally, the DALY estimation process in the Global Burden of Disease has evolved over the last two decades and is expected to continue changing in the coming years with the incorporation of new risk factors and diseases. However, the research enterprise outcome account is a static process. While the authors acknowledge the limitation of the DALY and mention that it's the best available health burden measure, did the authors consider some weighting method to mitigate the bias? For example, providing weight in accordance with the evolving nature of the DALY estimates.

R3.7. This is a question that came up during the development of our study as well. In a personal Email communication with Prof. Theo Vos (IHME) from earlier this year, he replied: "Estimates from each round of GBD are consistent in approach. Thus, the 1990 estimates from GBD2021 have been re-estimated based on all new data, methods changes." This means that within a given GBD release (such as GBD 2021, the data foundation of our study), the entire time series reflects a harmonized methodology. Nonetheless, we recognize that the evolving nature of DALY estimation can still introduce biases, especially for causes that have not been consistently tracked or defined across all GBD iterations. For this reason, we restricted our analysis to Level 2 disease categories, which are rather stable over time and consistently included in GBD reporting. This choice helps mitigate the risk of biased trends due to changing disease

definitions or category structures. We also considered your suggestion of applying a weighting scheme to account for the evolution of GBD reporting. However, given that the time series within each GBD round is already internally consistent, and that there is, to our knowledge, no transparent or validated method to quantify the degree of change attributable to estimation updates for specific causes or years, we found no empirical basis for constructing such weights without introducing further subjectivity. To strengthen the robustness of our findings, we complemented our DALY-based analysis with two additional outcome measures, mortality and prevalence, which are also provided by the GBD and subject to different modeling assumptions. While these measures are not immune to updates, they offer independent perspectives on disease burden and largely corroborate our main results. We have also added this point to our discussion of potential limitations inherent in the data foundation.

⇒ **Please see new Supplementary Figures 2A, 2B, and 2C, Discussion Section page 15, and revised Methods section page 26**

8. I'm not sure about the journal structure; currently, the introduction and results sections are difficult to separate. Providing the section title 'Results' before presenting the results may be helpful for readers.

R3.8. We have revised the structuring of the manuscript according to Nature Medicine requirements. Thank you.

Point-by-point response

Dear Professor Lerchenmueller,

Your revised Article, "Divergence between the distribution of research and disease burden" has now been seen by 3 referees. You will see from their comments that they find your revised manuscript strengthened and improved. In light of their positive feedback, and provided that you can address their remaining concerns as well as our editorial points below, we would be happy to offer to publish your study in Nature Medicine once this next round of revision is completed.

Please address the following editorial and formatting comments in the next round of revisions.

- The abstract has a word limit of strictly less than 200 words.
- Please define all abbreviations at first use and minimise the use of two letter abbreviations
- Please remove editorialising terms/language throughout the manuscript when describing the study and findings- i.e. "first", "novel", "largest/most comprehensive", "clearly", "large"... etc.
- Please provide a conclusion paragraph in the Discussion section
- Please be aware that we are unable to accommodate supplementary text. Please integrate them into either the main text or the methods section. Please note that there is no limit for our Methods section as it is online only.
- The supplementary material cannot include references. Please move the references in the supplementary material to the main reference list and ensure that this is appropriately cited in the main text.
- You currently have 7 main display items (6 figures, 1 table), and 8 supplementary figures. Please either merge two of your figures into a single figure or convert 1 of your main display items to extended display figures to enhance visibility (each of which must fit on a single page). Please note that we can permit 6 main display items and 10 Extended Data display items. Please convert some of your tables to Supplementary Tables, of which you can have an unlimited number.
- Please note that all Figures need to fit a single page (they can be multiple panels)
- The current Data Availability Statement is unacceptable. Please deposit the data into a public repository, or provide details regarding access of the data, including raw and aggregate data. We do not allow "data available upon request".
- The current Code Availability Statement is unacceptable. Please deposit the code into a public repository, and provide a functional weblink.
- You must provide a section on Competing Interests.
- Any references cited only in the methods needs to be included in a separate methods-only references section and should be numbered contiguously to the main reference list (i.e. number starts at XX following on from the numbering of the main reference list, not 1).

Thank you for offering to publish our manuscript in Nature Medicine. We have addressed all editorial points raised above. In particular:

- The abstract complies with the word limit, we now have 6 main display items, 9 extended data figures and a set of supplementary figures without additional supplementary text.
- All data have been deposited in a linked Figshare repository, and the Data Availability Statement and Code Availability Statement have been updated accordingly. The Figshare DOI will be generated upon manuscript submission and subsequently added to both statements.

In addition to addressing the remaining points from the reviewers, please edit your manuscript to comply with our formatting guidelines for Analyses, which are:

* Abstract: 200 words, unreferenced.

Completed.

* Main text: 4000 words with subheadings for the Introduction, Results and Discussion

We have reduced the text as much as possible without compromising the study's contribution. We have also added a more detailed exposition on the implications of the changing landscape of U.S. research and funding, which partially offset our reduction in text. We hope that the current length of the manuscript (~4,750 words) is acceptable.

* References: up to 60 in the main text + 20 methods-only references

Completed.

* Display items: up to 6 main display items (inclusive of figures and tables) and up to 10 Extended Data display items (inclusive of figures and tables). Extended Data are an integral part of the paper and only data that directly contribute to the main message should be presented.

Completed.

* Online Methods: no word limit; please provide the methods consolidated in a single section at the end of the main text document

Completed.

Reviewers' Comments:

Reviewer #1 (Remarks to the Author):

Thank you for the chance to see the revised version of this work. I do see many improvements in this updated manuscript, and I thank the authors for their revisions and answers to my comments/questions.

The authors seem to have addressed several of the concerns I raised well, including:

- language and framing that gives the impression of causality; glad to see this ameliorated
- Glad to see the new lagged analysis, and I find its results interesting
- Also happy to see more compelling Discussion of the results with some of the changes the authors made.
- Great that you also now address the changes in research funding in the US
- Thank you for adding the inter-rater metrics and validation for the LLM parsing of the affiliations, and kudos for now including affiliations of all authors
- Thank you for avoiding the normality assumption with DALY uncertainties and bootstrapped sampling

I have no further comments or questions for the authors.

Thank you for your comments throughout this review process.

Reviewer #2 (Remarks to the Author):

Thank you for the strong, detailed responses to each point I and the other reviewers made.

It was very interesting to read how you addressed the points and the changes made in the manuscript. I have no further comments.

Thank you for your comments throughout this review process.

Reviewer #3 (Remarks to the Author):

I appreciate the authors for updating the manuscript accordingly. I'm happy with the current version to be accepted for publication in Nature Medicine.

Thank you for your comments throughout this review process.